# Unusual flexibility of transparent poly(methylsilsesquioxane) aerogels by surfactant-induced mesoscopic fiber-like assembly

Ryota Ueoka[1,5], Yosuke Hara[1,5], Ayaka Maeno[2], Hironori Kaji [2], Kazuki Nakanishi [3,4] & Kazuyoshi Kanamori [1] ✉

High-performance thermal insulators represented by aerogels are regarded as one of the most promising materials for energy savings. However, significantly low mechanical strength has been a barrier for aerogels to be utilized in various social domains such as houses, buildings, and industrial plants. Here, we report a synthetic strategy to realize highly transparent aerogels with unusually high bending flexibility based on poly(methylsilsesquioxane) (PMSQ) network. We have constructed mesoscopic fine fiber-like structures of various sizes in PMSQ gels by the combination of phase separation suppression by tetramethylammonium hydroxide (TMAOH) and mesoscopic fiber-like assembly by nonionic poly(ethylene oxide)-*b*-poly(propylene oxide)-*b*-poly(ethylene oxide) (PEO-*b*-PPO-*b*-PEO) type surfactant. The optimized mesoscale structures of PMSQ gels have realized highly transparent and resilient monolithic aerogels with much high bendability compared to those reported in previous works. This work will provide a way to highly insulating materials with glasslike transparency and high mechanical flexibility.

Transparent aerogels, which show low thermal conductivity (down to 12 mW m⁻¹ K⁻¹) with low bulk density (typically 0.001–0.2 g cm⁻³), are sol–gel materials with a mesoscale homogeneous porous structure[1,2]. These unique features of visible transparency and low thermal conductivity make aerogels an ideal material for highly insulating devices including windows for efficient energy savings[3–5]. Aerogels are, however, mechanically weak materials due to their weakly linked skeletons and high porosity. To improve their mechanical strength, many efforts have been devoted by controlling the molecular-scale network[6] and forming composite with mechanically more reliable materials such as organic polymers and glass fibers[7–9]. However, such structural controls and modifications while maintaining visible transparency have been

challenging due to the lack of knowledge in structurization in the mesoscale in high-porosity materials.

Fundamental studies on influences of the macroscale (e.g., in micrometer scales) porous structure on their mechanical properties have been intensively studied through experimental and simulation approaches, in which it has been found architectural features, in addition to the chemical structure of the network, influence their mechanical strength and flexibility[10–12]. When looking at the mesoscale (e.g., in nanometer scales) porous structure in aerogels, the fundamental knowledge of the relationship between the porous structure and their mechanical properties would give an opportunity to realize highly flexible aerogels through the control over the mesoscale porous

[1]Department of Chemistry, Graduate School of Science, Kyoto University, Kitashirakawa, Sakyo-ku, Kyoto 606-8502, Japan. [2]Institute for Chemical Research, Kyoto University, Gokasho, Uji, Kyoto 611-0011, Japan. [3]Institute of Materials and Systems for Sustainability, Nagoya University, Furo-cho, Chikusa-ku, Nagoya, Aichi 464-8601, Japan. [4]Institute for Integrated Cell-Material Sciences, Kyoto University, Yoshida, Sakyo-ku, Kyoto 606-8501, Japan. [5]These authors contributed equally: Ryota Ueoka, Yosuke Hara. ✉e-mail: kanamori@kuchem.kyoto-u.ac.jp

structure. In particular, three-dimensional (3D) branched fibrous architecture can be considered as a promising way to realize high flexibility due to the absence of the "necks" where stress would be accumulated. As demonstrated in nanofiber-based aerogels such as cellulose/chitosan nanofiber aerogels and carbon nanotube (CNT) aerogels, designing a 3D fibrous structure brings out high flexibility in aerogels[13–15]. However, the optimized mesoscale fibrous structure which realizes both high flexibility and transparency is still unclear due to the difficulty in designing the mesoscale fibrous structure built up with a well-controlled molecular-scale network.

Here, we demonstrate a synthetic strategy to realize highly transparent aerogels with high bending flexibility through the optimization of the mesoscale porous structure by surfactant-induced mesoscopic fiber-like assembly. We employed poly(methylsilsesquioxane) (PMSQ, $CH_3SiO_{3/2}$) aerogels[16,17] as an example. Transparent PMSQ aerogels have been prepared by hydrolysis and polycondensation of methyltrimethoxysilane (MTMS, $CH_3Si(OCH_3)_3$), in the presence of surfactant as a phase separation suppressor. Formally, the PMSQ network can be prepared by the following two-step acid-base process in these examples:

1. Acid-catalyzed hydrolysis

$$CH_3Si(OCH_3)_3 + 3H_2O \rightarrow CH_3Si(OH)_3 + 3CH_3OH$$

2. Base-catalyzed polycondensation

$$CH_3Si(OH)_3 \rightarrow CH_3SiO_{3/2} + 3/2H_2O$$

As seen in the PMSQ aerogels, some research has been demonstrated the mechanical flexibility, such as spring-back behavior, of the organic-inorganic hybrid aerogels with some transparency derived from the organic-inorganic hybrid networks with high hydrophobicity[6,16,18]. However, to the best of our knowledge, there are no reports on strategic syntheses to optimize the mesoscale structure of aerogels for both high transparency and high mechanical flexibility, especially bending flexibility. The key to our approach toward fiber-like structured PMSQ aerogels is a use of nonionic surfactant PEO-b-PPO-b-PEO (known as Pluronic) as the structure determining agent. According to the literature[19,20], the hydrophobic network of PMSQ showed high tendency of macroscopic phase separation (spinodal decomposition) in aqueous media. Then, in the case of Pluronic F127 (properties of Pluronic surfactants are listed in Supplementary Table 1), for example, a sol–gel system with high concentrations (e.g., $0.1\,g\,mL^{-1}$ or more) of the surfactant F127 was found to moderately suppress the macroscopic phase separation and resulted in translucent ($T_{550} < 60\%$, transmittance at 550 nm through a 10-mm-thick specimen) PMSQ aerogels[16,21]. Although no aerogels with high transparency (>90%) have been reported in that system, those PMSQ aerogels appear to have a somewhat long skeleton structure unlike the typical pearl-necklace-like skeleton network. Inspired by these results, we came up with an idea that this structural diversity of PMSQ gels prepared in the presence of the Pluronic-type surfactant would provide an opportunity to obtain highly transparent PMSQ aerogels with fine and well-defined branched fiber-like structures in the mesoscale.

In this work, as a proof of concept, we tried to control the mesoscopic assembly of the PMSQ network by employing several kinds of the Pluronic-type triblock copolymer both as phase separation suppressors and structural determining agents. In addition, an organic strong base, tetramethylammonium hydroxide (TMAOH), was employed as a polycondensation catalyst, which is effective for equilibration of polysiloxanes[22,23]. By identifying and optimizing the synthesis parameters such as starting composition, PMSQ aerogels with both high transparency and high bending flexibility have been demonstrated.

## Results
### Identifying appropriate synthetic parameters of the PMSQ aerogels with fiber-like structure
The typical procedure to prepare the PMSQ aerogels is as follows: (1) MTMS was hydrolyzed in 5 mM acetic acid; (2) a given amount of surfactant and water were subsequently added to the solution to obtain the homogeneous sol; (3) a given amount of aqueous TMAOH was added into the sol at 0 °C for polycondensation of hydrolyzed MTMS to form a wet gel; and (4) the obtained gel was washed with water, methanol, and 2-propanol in sequence, and then the washed alcogel was supercritically dried from carbon dioxide at 80 °C under 14 MPa.

The microstructure of the PMSQ aerogels depends mainly on the ratio of surfactant to MTMS and the concentrations of surfactant and MTMS in the starting solution. In our previous studies, several kinds of nonionic[24] and cationic surfactants[25] have been shown to suppress the macroscopic phase separation to lead to the mesoporous structure. Supplementary Figs. 1 and 2 demonstrate that the finer porous structure is obtained with increasing amount of F127 and n-hexadecyltrimethylammonium chloride (CTAC), respectively. Without surfactants, a high phase separation tendency between hydrophobic PMSQ polymer and water-based solvent resulted in the macroporous structure with spheroidized skeletons[26] (Supplementary Fig. 1a). The transient co-continuous macroporous structure of phase separation, which was solidified by the sol–gel transition, has been observed in the case of F127 as phase separation suppressor (Supplementary Fig. 1b, c). On the other hand, when CTAC was employed as the phase separation suppressor, the transient structure of spinodal decomposition has not been observed, and pore skeletons with aggregated particles have been observed instead (Supplementary Fig. 2b).

To identify the suitable synthetic process which realizes both high transparency and high flexibility, we explored the starting composition to form the mesoscale branched fiber-like structure by using several kinds of PEO-b-PPO-b-PEO-type triblock copolymers. As mentioned above, the co-continuous structure, which are derived from the spinodal decomposition in the micrometer scale, were formed in the presence of F127, while the presence of CTAC resulted in the macroporous structure with aggregated particles in the case of moderate phase separation tendency. We assumed that the mesoscale branched fiber-like structure would be isotopically extended with a lower concentration of MTMS. We therefore decreased the concentration of MTMS from 5 mL (35 mmol) in 7 mL of water to 5 mL in 12 mL of the same.

The phase separation tendency between MTMS-derived methylsilsesquioxane (MSQ) condensates and water-based solvent is dominated mainly by the concentration and kinds of base catalyst and surfactant, which influences the molecular weight of MSQ and affinity between the gelling phase and the solvent-based phase. First, we discuss the effect of base catalyst. Supplementary Figure 3 shows the influences of the base catalyst on the transparency of the PMSQ aerogels in the system containing surfactant F127. In the case of urea used in the previous studies, hydrolysis of urea generates ammonia as the polycondensation catalyst, and the solution pH gradually increases to 7 – 9. In the case of TMAOH, the solution pH rapidly increases to higher values (from ~12 (0.010 M) to ~14.3 (2.0 M)) because a strong base TMAOH is separately added after the hydrolysis of MTMS. The obtained PMSQ aerogels prepared with TMAOH had higher transparency (except for the case of 0.010 M TMAOH) than that of the PMSQ aerogel prepared with urea as the base source. As discussed in the silicate system, it is reasonable to speculate that TMAOH more effectively suppresses the phase separation tendency, because the MSQ species ionically bound with the $TMA^+$ cations increase their affinity in the water-methanol solution[27,28]. By varying the concentration of TMAOH, transparency of the PMSQ aerogels has been controlled and shown maximum (83%) at 0.50 M (Supplementary Fig. 3). The

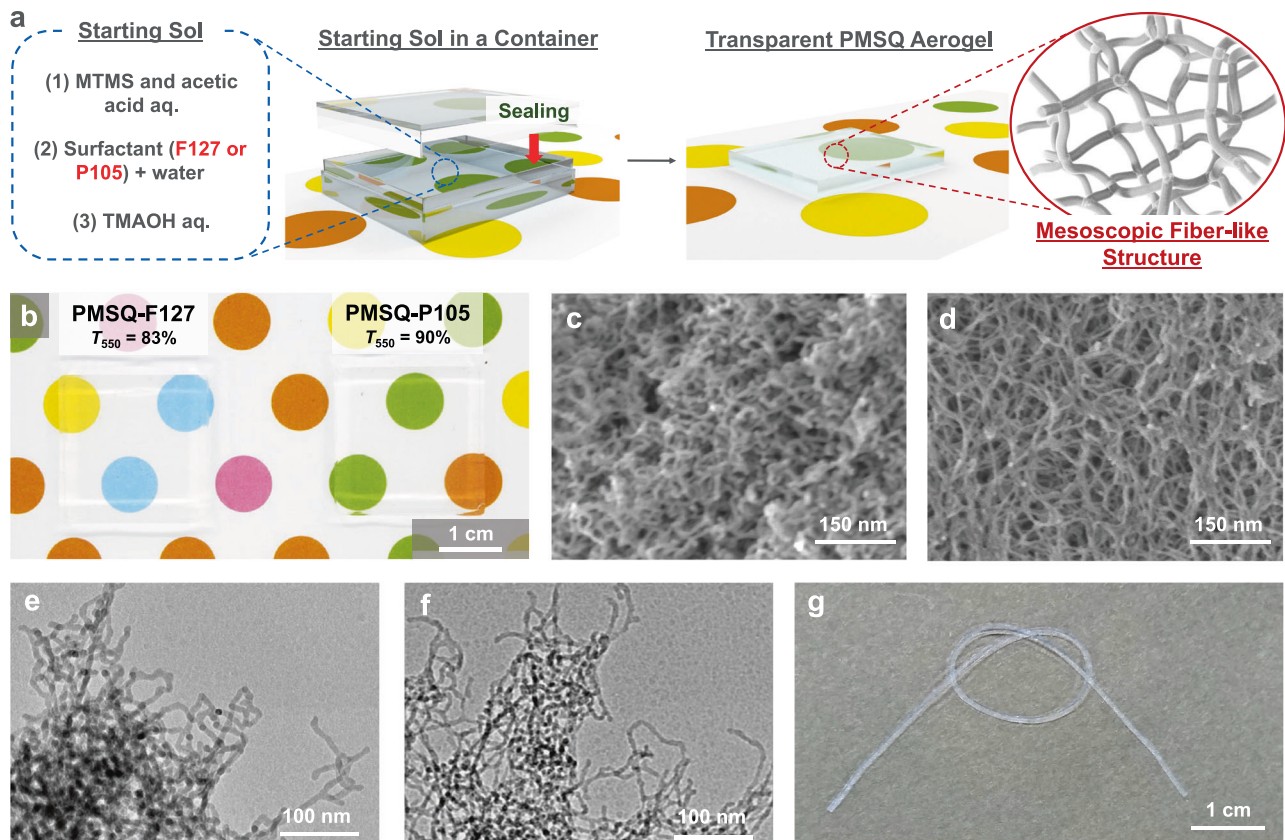

**Fig. 1 | Macro-/mesoscopic physical features of the flexible and transparent PMSQ aerogels prepared with two types of surfactants. a** The starting compositions were optimized to prepare highly transparent PMSQ aerogels with the mesoscale fibrous structure. **b** Appearance of the PMSQ aerogels prepared with the different surfactant, F127 and P105. **c, d** FE-SEM images of the samples PMSQ-F127 and PMSQ-P105, respectively. **e, f** TEM images of the samples PMSQ-F127 and PMSQ-P105, respectively. **g** A photograph of the string-shaped sample of PMSQ-F127, demonstrating high flexibility like a silicone tube.

field-emission scanning electron microscopy (FE-SEM) images in Supplementary Fig. 4 show that the porous structure is composed of branched one-dimensionally long skeletons of ~10 nm thickness and becomes finer with the increasing concentration of TMAOH.

In addition to the optimization of the TMAOH concentration, we identified the appropriate amount of the surfactant toward the maximization of transparency. The details of the starting compositions are summarized in Supplementary Table 2. We first focus on two types of PEO-*b*-PPO-*b*-PEO surfactant, F127 and P105. The molecular structures of F127 and P105 are expressed as $EO_{106}PO_{70}EO_{106}$ ($M_w$ ~ 12,600) and $EO_{37}PO_{56}EO_{37}$ ($M_w$ ~ 6500), respectively (Supplementary Table 1)[29]. The hydrophile-lipophile balance (HLB) values of F127 and P105 are 18 – 23 and 12 – 18, respectively.

**Physical features of the PMSQ aerogels with fiber-like structure**
Figure 1 shows macro-/mesoscopic physical features of the PMSQ aerogels with the mesoscale branched fiber-like structure prepared in the presence of F127 and P105. A schematic of the synthesis of the PMSQ aerogels is shown in Fig. 1a. In the optimized starting compositions to maximize the transparency, light transmittance at 550 nm through a 10-mm equivalent specimen ($T_{550}$) of the PMSQ aerogels (denoted as PMSQ-F127 and PMSQ-P105) reached 83% and 90%, respectively (Fig. 1b). These values are among the highest compared to the PMSQ aerogels reported in previous works[24,25]. The skeletal features observed by FE-SEM and transmission electron microscopy (TEM) of PMSQ-F127 and PMSQ-P105 are shown in Fig. 1c, d and Fig. 1e, f, respectively. The pore structures are constructed with long one-dimensional sections: fiber-like skeletons of ~6–8 nm thickness with

branches (nodes). The size parameters of the structure are discussed later. Because the material shape of the PMSQ aerogels depends on the shape of the vessel used in the sol–gel synthesis, the PMSQ aerogels can be prepared in different forms; for examples, plate (Fig. 1b) and string (Fig. 1g). Taking the PMSQ-F127 aerogel as an example, one unique feature we found in the PMSQ aerogels with the branched fiber-like skeletons is the extraordinarily high flexibility like a silicone wire or tube as shown in Fig. 1g.

**Mechanical properties of the PMSQ aerogels with fiber-like structure**
Three-point bending test results of PMSQ-F127 and PMSQ-P105 are shown in Fig. 2. Bulk density ($\rho_b$) is 0.13 g cm⁻³ for PMSQ-F127, and 0.12 g cm⁻³ for PMSQ-P105. It should be emphasized here that the monolithic samples in a cylindrical shape show large bending deformations. Some studies have reported that thin aerogel samples have superior bending deformability[13,14,30,31]. However, thin materials such as glass sheets can be more bendable compared to bulky counterparts because of the decreased section modulus. Although there is no standard for three-point bending test of aerogels, it is often preferred to use a short span length of about 20 mm for thick specimens[32–35]. Therefore, three-point bending tests with 20 mm span were first performed on thick cylindrical aerogel specimens of about 10 mm in diameter (Fig. 2a, b). The maximum bending strain at failure with a 20 mm span ($\varepsilon_{max,20}$) is 51% and 75% for PMSQ-F127 and -P105, respectively (Fig. 2a). As will be discussed in more detail later, the PMSQ aerogel with comparable high transparency and low bulk density obtained in the previous work[25] (PMSQ-prev) has an $\varepsilon_{max,20}$ of 30%,

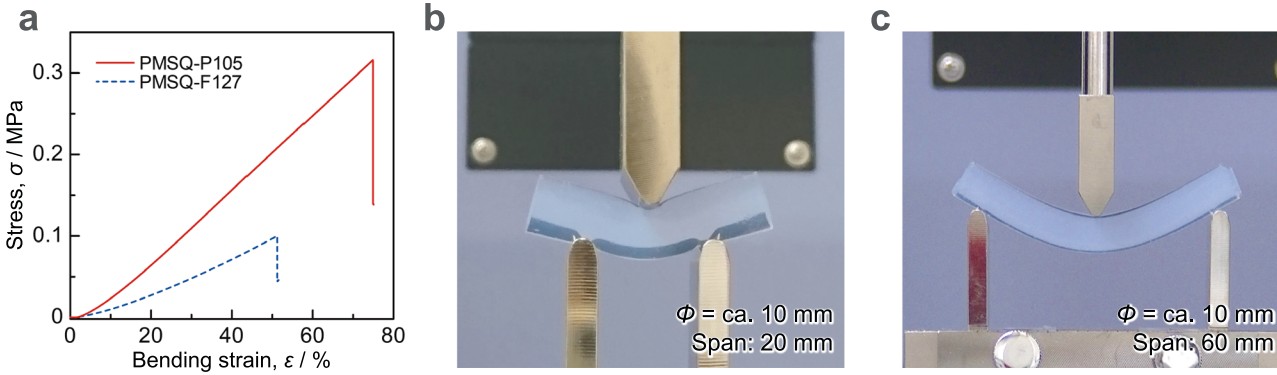

**Fig. 2 | Mechanical properties for bending deformation of the flexible and transparent PMSQ aerogels prepared in different conditions. a** Stress–strain curves obtained from three-point bending of the PMSQ aerogel samples with 20 mm span. **b**, **c** Photographs showing the bendability of (**b**) PMSQ-P105 (span: 20 mm) and (**c**) PMSQ-F127 (span: 60 mm). Both photos show the aerogels just before the failure.

**Table 1 | Comparison of the properties of the aerogels obtained in this work with those of reported aerogels**

| Sample | $\rho_b{}^*$ /g cm$^{-3}$ | Optical transparency† | $\varepsilon_{max}{}^\ddagger$ /% | Reference |
|---|---|---|---|---|
| PMSQ-F127 | 0.13 | Transparent (83%) | 51 | This work |
| PMSQ-P105 | 0.12 | Transparent (90%) | 75 | This work |
| Polymer-crosslinked silica | 0.63 | Opaque | 40 | [32] |
| Hexylene-bridged poly(silsesquioxane) | 0.093 | Opaque | 40 | [33] |
| Surface-modified hexylene-bridged poly(silsesquioxane) | 0.22 | Transparent (56%) | 38 | [34] |
| Silica | 0.20 | Not shown | 8 | [34] |
| POSS-based polysiloxane | 0.20–0.21 | Opaque | 16–18 | [35] |

*Bulk density.

†Estimated from the appearance of the aerogels. The light transmittance values at 550 nm through 10-mm equivalent specimen are shown in the parentheses.

‡The maximum strain at failure estimated from stress–strain curves obtained from three-point bending of aerogels.

indicating that PMSQ-F127 and -P105 can be bent even higher. Table 1 shows comparisons with other aerogels that have shown comparable three-point bending behaviors to this work. Note that it is difficult to make completely accurate comparison with other aerogels obtained in previous studies because the results of three-point bending tests are strongly influenced by the testing condition and sample geometry. However, focusing only on bending flexibility, there is no aerogel that has as high bending flexibility as PMSQ-F127 or -P105. In addition, only silica aerogels have comparable high transparency to PMSQ-F127 or -P105 and other non-silica aerogels have significantly lower transparency.

It is worth noting that the bending test with a short span does not exhibit natural curvature in thick specimens; i.e., an accurate evaluation of bending performance has not been done[36]. We therefore have also performed three-point bending tests with a long span length (60 mm). In this testing condition, PMSQ-F127 shows large bending deformation (Fig. 2c), which is almost the same as PMSQ-P105. The obtained stress–strain curves are shown later. Supplementary Movie 1 and 2 demonstrate the high bendability of PMSQ-P105 and -F127, respectively. The stress–strain recovery behavior of a cyclic three-point bending test on PMSQ-F127 is shown in Supplementary Fig. 5. The specimen was bent just before breaking (Supplementary Fig. 5b), and when the load was removed, it returned to a slightly deformed state (Supplementary Fig. 5c). After this test, the specimen was heated at 120 °C overnight, but did not show a complete recovery. Similarly, residual strains were observed on specimens of PMSQ-F127 and -P105 after 50% uniaxial compression test (Supplementary Fig. 6). In this case, however, the specimens recover nearly to their original state by heating at 110 °C for a few hours (Supplementary Fig. 7). These results indicate that both PMSQ-F127 and -P105 are viscoelastic.

## Insights into the origin of the mechanical flexibility of the PMSQ aerogels with fiber-like structure

The mechanical properties of low-density porous materials are naturally influenced by the shape of pore skeletons and the properties of the solid constituent as deeply studied in the cellular solids[10,37,38]. To better understand the high bending flexibility of PMSQ-F127 and -P105, therefore, we investigated the effect of the pore and molecular structures. As controls, we prepared two types of PMSQ aerogels with different pore structures while controlling the following properties: bulk density, transparency, and material shape (Fig. 3a). Bulk density is related to porosity, which influences the mechanical properties[39,40], while transparency is related to the size and homogeneity of the porous structure. One is a PMSQ aerogel prepared with a different cationic surfactant CTAC (denoted as PMSQ-prev). This is the same as reported in our previous work[25] (see Methods for details). The other is a PMSQ aerogel prepared with the same Pluronic P105 and *N, N*-dimethylformamide (DMF) as a part of the solvent (denoted as PMSQ-P105DMF). Since the polarity of DMF is lower than that of water and specific hydrogen bonding can be formed with silanol groups[41], the formation of the solid phase of the MTMS-derived hydrophobic condensates is varied in the additional presence of DMF. Both PMSQ-prev and -P105DMF have fine pore structures, that are similar to the pore structure of a standard silica aerogel with mass fractals[42,43] (Supplementary Fig. 8). Figure 3b shows the stress–strain curves obtained from three-point bending tests of PMSQ-F127, -P105, -prev, and -P105DMF. Obviously, PMSQ-F127 and -P105 have higher maximum bending strains at failure ($\varepsilon_{max,20}$ = 51% and 75%, respectively) than the other two ($\varepsilon_{max,20}$ = 30% for PMSQ-prev and 24% for -P105DMF).

To clarify differences in pore structure, we introduce the concept of skeletal ratio (Fig. 3c). The pore skeletons can be classified into (i) 1D skeletons and (ii) their connections (nodes). The average length and

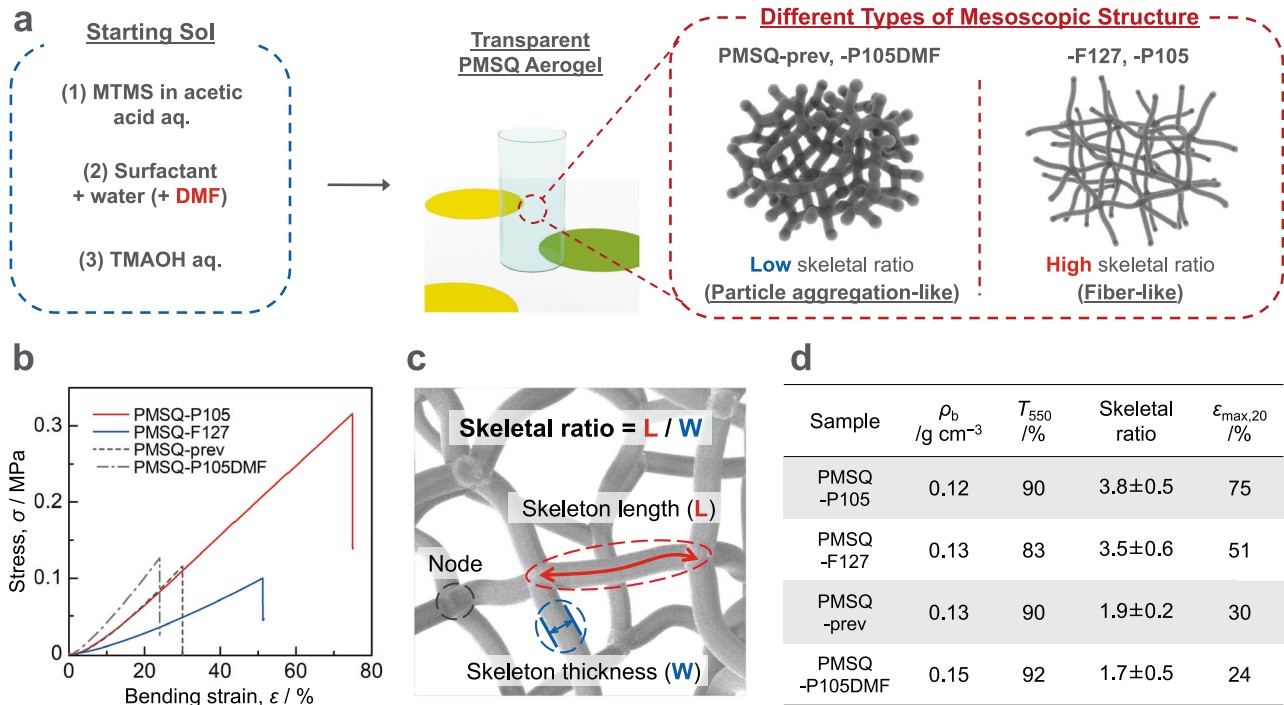

**Fig. 3 | Control of the mesoscale structure of the PMSQ aerogel, and properties of the PMSQ aerogel samples with different mesoscale structures. a** A schematic of controlling the mesoscale structure. **b** Stress–strain curves obtained from three-point bending of the PMSQ aerogel samples with 20 mm span. **c** A schematic of the pore skeletons with nodes. Skeletal ratio is defined as [skeleton length]/[skeleton thickness]. **d** Bulk density ($\rho_b$), light transmittance at 550 nm through 10- mm equivalent specimen ($T_{550}$), skeletal ratio calculated by using the means with a 95% confidence interval of [skeleton length] and [skeleton thickness] from FE-SEM images (the error shows the estimated maximum error), and maximum strain at failure of the three-point bending ($\varepsilon_{max,60}$) at 20 mm span of the PMSQ aerogel samples.

thickness of the branched 1D skeletons are denoted as skeleton length and skeleton thickness, respectively. The skeletal ratio is defined as (skeleton length)/(skeleton thickness). A high skeletal ratio gives the appearance of a fiber-like structure, while a low skeletal ratio gives the appearance of a typical particle aggregation-like structure (Fig. 3a). We calculated some kinds of size parameters, including skeletal ratio, from high-resolution FE-SEM and TEM images (see Supplementary Table 3 and Supplementary Fig. 15 for details). PMSQ-F127 and -P105 have higher skeletal ratios ($3.5 \pm 0.6$ and $3.8 \pm 0.5$, respectively) than those of PMSQ-prev and -P105DMF ($1.9 \pm 0.2$ and $1.7 \pm 0.5$, respectively), while these four samples have similar bulk density and transparency (Fig. 3d).

Supplementary Figure 9 shows the comparison of solid-state $^{29}$Si cross-polarization magic angle spinning (CP/MAS) nuclear magnetic resonance (NMR) spectra of the PMSQ aerogels (-F127, -P105, -prev, and -P105DMF). Note that we have confirmed that there is no obvious difference between CP/MAS and single-pulse measurements in the PMSQ system. Peaks around −67 ppm and −57 ppm correspond to fully condensed ($CH_3Si(OSi)_3$, $T^3$) and doubly condensed ($CH_3Si(OSi)_2(OH/CH_3)$, $T^2$) silicon species, respectively. Although there are only negligible differences in the peak shape among these aerogels, the condensation degree values, $(T^3 + 2/3T^2)/(T^3 + T^2)$, calculated from the peak areas, are slightly different. The condensation degree is 97.4%, 98.3%, 97.7%, and 97.6% for PMSQ-F127, -P105, -prev, and -P105DMF systems, respectively. In general, from the point of view of molecular-level structures, the lower condensation degree or cross-linking density should result in lower modulus and higher bendability. In the case of trifunctional MTMS, elasticity or resilience may be sacrificed due to the more residual alkoxy/hydroxy groups in the less cross-linked network. In the present case, however, there is no clear correlation between the bending flexibility and cross-linking density: PMSQ-P105 is the most bendable and has the highest cross-linking density, but the

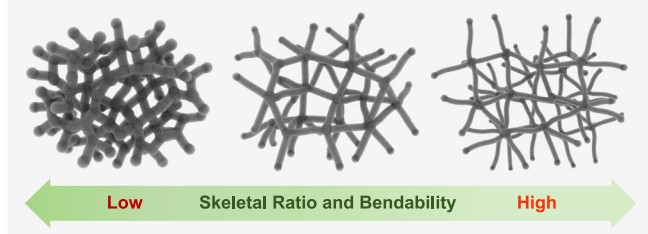

**Fig. 4 | A Schematic of the relationship between the mesoscale pore structures and bulk mechanical properties of aerogels.** The higher the skeletal ratio is, the more fiber-like the porous structure looks, and accordingly, the higher the bending deformability of the monolithic aerogel is. Note that all of the monolithic aerogels with these structures have the same bulk density.

second most bendable aerogel, PMSQ-F127, has the lowest cross-linking density; also, both PMSQ-F127 and -prev have almost the same $^{29}$Si CP/MAS NMR spectra and cross-linking density, but PMSQ-F127 can be bent much higher than PMSQ-prev. In addition, no remaining surfactant, which may influence the mechanical properties[44], was detected in all the samples including PMSQ-P105 and -P105DMF by $^{13}$C NMR and TG-DTA measurements as shown in Supplementary Fig. 10a and 10b, respectively. However, the bending deformability of PMSQ-P105 and -P105DMF is significantly different even though the same surfactant is used in their preparation. Based on the above results, we conclude that the mesoscale branched fiber-like porous structure as seen in PMSQ-F127 and -P105 provides a considerable improvement in the bending flexibility.

The possible relationship between the skeletal ratio and the bending flexibility of aerogels is shown in Fig. 4. Just as glass fibers can be bent a lot, a branched 1D skeleton with a higher skeletal ratio can be

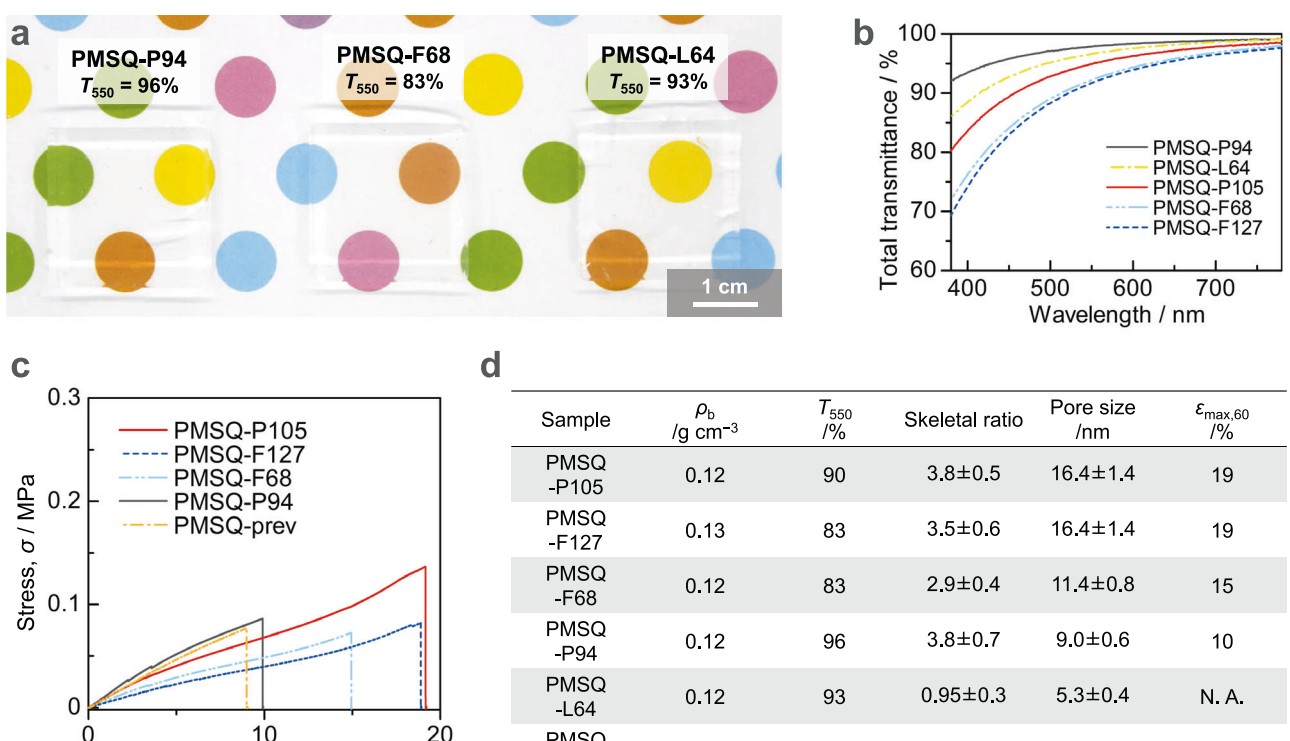

**Fig. 5 | Influence of the kind of Pluronic-type surfactant on optical transparency and mechanical properties.** The stating compositions are optimized to prepare highly transparent PMSQ aerogels. **a** Appearance of the PMSQ aerogels prepared with the different surfactants: P94 (left), F68 (center), and L64 (right). **b** Spectra of the total transmittance of the PMSQ aerogel samples. The thickness of the samples is 5.0 mm. **c** Stress–strain curves obtained from three-point bending of the PMSQ aerogel samples with 60 mm span. **d** Bulk density ($\rho_b$), light transmittance at 550 nm through 10-mm equivalent specimen ($T_{550}$), skeletal ratio calculated by using the means with a 95% confidence interval of [skeleton length] and [skeleton thickness] from FE-SEM images (the error shows the estimated maximum error), mean pore size with a 95% confidence interval calculated from FE-SEM images, and maximum strain at failure of the three-point bending ($\varepsilon_{max,60}$) at 60 mm span of the PMSQ aerogel samples.

bent more than that with a lower skeletal ratio. For random cellular architectures such as aerogels, it is well known that the effective properties are dominated by bending deformation of the ligaments[10,37,38]. Therefore, when there is no significant difference in the density and transparency, that is, the size and homogeneity of the pore structures, of aerogels, the higher bendability of branched 1D skeletons with a higher skeletal ratio would contribute to the higher bendability of the aerogels. As will be discussed later, however, even if the skeletal ratio is high, it is possible for aerogels to break before fully bending if the cell size (pore size and skeleton thickness) is too small.

**Optimization of the mesoscale porous structure of PMSQ aerogels toward high thermal insulation materials with both high transparency and high flexibility**

To demonstrate the optimization of mesoscale structure toward high thermal insulation materials with both high transparency and high flexibility, we tried to identify the suitable starting composition using other PEO-*b*-PPO-*b*-PEO-type triblock copolymers as phase separation suppressor and structural determining agent (Supplementary Table 1, 2).

We employed three other types of Pluronic surfactants: F68 (the hydrophilic-lipophilic balance (HLB) value > 24 and $M_w \sim 8400$), P94 (HLB value 13.5 and $M_w \sim 5000$), and L64 (HLB value 12–18 and $M_w \sim 2900$). The properties of the optimized PMSQ aerogels obtained using these Pluronic-type surfactants (denoted as PMSQ-F68, PMSQ-P94, and PMSQ-L64, respectively) are shown in Fig. 5, Supplementary Fig. 11 (FE-SEM and TEM images), and Supplementary Fig. 12 (nitrogen adsorption-desorption isotherms and BJH pore size distributions). Note that the results of nitrogen adsorption-desorption measurements

should not reflect the correct pore structures of the aerogels due to sample shrinkage caused by the capillary pressure of adsorbing nitrogen during the measurements[45,46]. Therefore, some size parameters calculated from FE-SEM and TEM images for these samples are also used for better discussion (see Supplementary Table 3 for details). When using surfactant F68, which has a molecular weight ($M_w \sim 8400$) between that of F127 ($M_w \sim 12,600$) and that of P105 ($M_w \sim 6500$), obtained PMSQ-F68 has almost the same transparency as PMSQ-F127 and slightly lower bending deformability than PMSQ-F127 and -P105 (Fig. 5a–c and Supplementary Fig. 13). While the pore size of PMSQ-F68 is slightly smaller than those of PMSQ-F127 and -P105, the pore size distribution is slightly broader in terms of the full width at half-maximum, which may result in transparency similar to PMSQ-F127 (Fig. 5d, Supplementary Fig. 11a, and Supplementary Fig. 12). The pore framework of PMSQ-F68 has long one-dimensional sections similar to -F127 and -P105 (Supplementary Fig. 11d), and the value of the skeletal ratio is $2.9 \pm 0.4$, which is lower than that of -F127 and -P105 ($3.5 \pm 0.6$ and $3.8 \pm 0.5$, respectively) but higher than that of -prev ($1.9 \pm 0.2$), which would lead to the medium bendability (Fig. 5c, d and Supplementary Table 3). When using surfactant P94 or L64, both of which have similar HLB values and lower molecular weight than that of P105, obtained PMSQ aerogels are more transparent than PMSQ-P105 (Fig. 5a, b). The surfactant L64 has the lowest molecular weight of the Pluronic-type surfactants used in this study, and PMSQ-L64 has the smallest pore size (Supplementary Fig. 11c and 12). The calculated mean pore size of PMSQ-L64 is also the smallest among the PMSQ aerogels ($5.3 \pm 0.4$), and the skeletal ratio value is $0.95 \pm 0.3$, which is only about half that of PMSQ-prev, $1.9 \pm 0.2$ (Fig. 5d). Presumably due to this too fine pore structure, PMSQ-L64 is too fragile to obtain a crack-free monolith. It

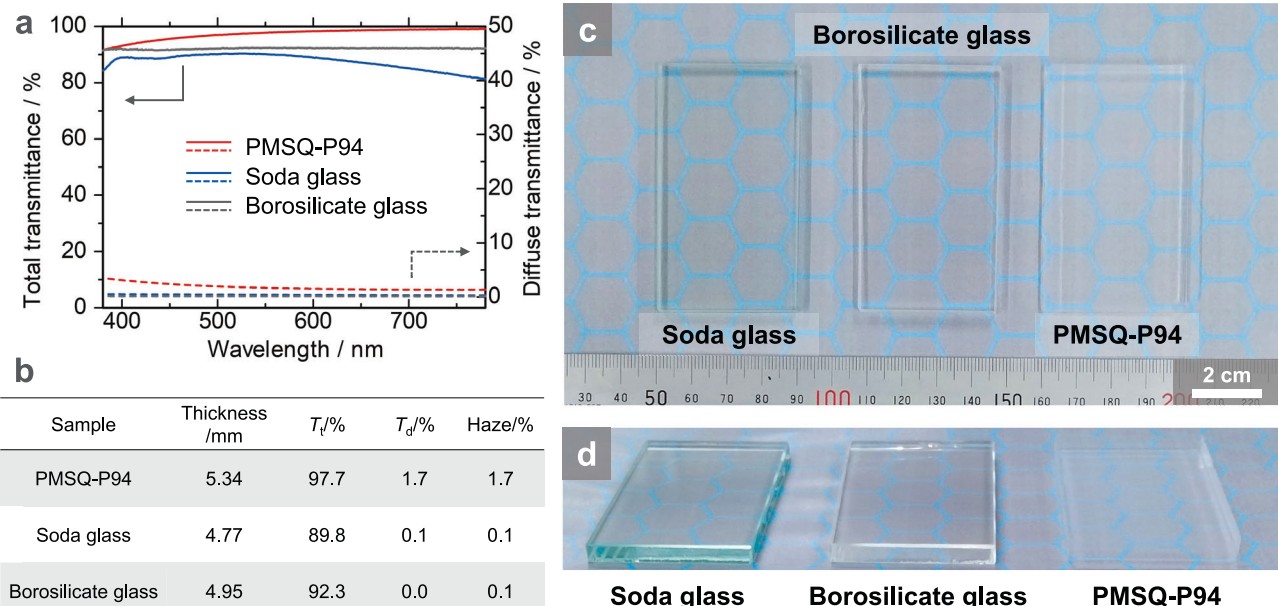

**Fig. 6 | Comparison of optical properties between the most transparent PMSQ aerogel obtained in this study and the conventional glass plates. a** Spectra of the total (solid lines) and diffuse (broken lines) transmittance of the PMSQ aerogel and glass plates. **b** Total transmittance ($T_t$), diffuse transmittance ($T_d$), and haze ($T_d/T_t \times 100$) of the samples. All of these data were measured with an integrating sphere in the visible light range (380–780 nm). (**c**, **d**) Photographs of the PMSQ aerogel and glass plate specimens. These samples were viewed (**c**) from above and (**d**) at an angle. The thickness is 4.8 mm (for soda glass), 5.0 mm (for borosilicate glass) and 5.3 mm (for PMSQ-P94).

was therefore difficult to perform mechanical tests on PMSQ-L64. On the other hand, PMSQ-P94 is tough and can be bent slightly higher than PMSQ-prev even though the pore size of PMSQ-P94 is the second smallest, close to that of PMSQ-L64 (Fig. 5, Supplementary Fig. 11 and 12). This is presumably because PMSQ-P94 has a high skeletal ratio ($3.8 \pm 0.7$), unlike PMSQ-L64 (Fig. 5d). The reason why PMSQ-P94 cannot be bent as high as PMSQ-F127 or -P105 is probably due to its too small pores and skeletons. In summary, both PMSQ-P94 and -L64 have similar small pore sizes, which lead to very high transparency, but their skeletal ratios are different; PMSQ-L64, which has a low skeletal ratio, is very friable because its porous structure can hardly be deformed, while PMSQ-P94, which has a high skeletal ratio, is tough and can be bent more because its porous structure can be deformed to some extent.

As mentioned above, PMSQ-P94 is tough despite its very high transparency ($T_{550} = 96\%$), so it has the potential to be used as a thermally insulating window material inserted in the glass pane if the size scale can be increased. Figure 6 shows the optical properties of PMSQ-P94 compared to conventional non-thermally insulating glasses with comparable thickness. Notably, the total transmittance of PMSQ-P94 in the visible light range (380–780 nm) is higher than those of the conventional glass plates (Fig. 6a, b). However, PMSQ-P94 has a small amount of diffuse transmittance, which contributes to slight haze (Fig. 6a, b). The haze value of PMSQ-P94 is 1.7% at 5.34 mm thickness, not zero like glass (Fig. 6b), which is comparable with the lowest value ever reported[47], and PMSQ-P94 is one of the most transparent aerogels to the best of our knowledge[47,48]. Because of this good optical property, it is difficult to see the difference between PMSQ-P94 and glass in terms of transparency when viewed from the above (Fig. 6c). The low, but not zero, haze value of PMSQ-P94 is primarily due to the absence of aggregation in the scale of ~100 nm or larger and to the surface roughness. The Rayleigh scattering can be suppressed by the skeletons with small mean particle size[47,49,50]. The pore structure of PMSQ-P94 is very fine (Supplementary Fig. 11b, e), and the size of the pore skeletons are very small as shown in Supplementary Table 3. On the other hand, when the plate-like monolithic samples are viewed from an angle, the

glass plates appear glossy, but the aerogel PMSQ-P94 does not, which indicates that the surface of PMSQ-P94 is rough to some degree (Fig. 6d).

Because of the toughness, PMSQ-P94 was able to be prepared in a larger scale, and then the thermal conductivity was measured. The thermal conductivity of PMSQ-P94 is 14.5 mW m⁻¹ K⁻¹, which is comparable with that of conventional silica aerogels and low enough for highly insulating window applications. Therefore, as demonstrated in Supplementary Fig. 14, the highly transparent and thermally insulating aerogel, PMSQ-P94, has good potential to be used as a window material because we can clearly see what is behind the aerogel. However, further reduction of the haze will be needed, which could be improved by increasing the surface smoothness of the aerogel.

## Discussion

Focusing on PEO-*b*-PPO-*b*-PEO-type surfactants, the molecular weight and HLB value can affect the pore skeleton structure and the transparency of the resulting PMSQ aerogel. When using a PEO-*b*-PPO-*b*-PEO-type surfactant with high molecular weight, e.g., Pluronic F127 ($M_w$ ~ 12,600) and P105 ($M_w$ ~ 6500), the pore skeleton tends to be one dimensionally long (mesoscopic fiber-like structure with high skeletal ratio), which would be beneficial for bulk bendability (Fig. 4). On the other hand, one with low molecular weight, e.g., Pluronic L64 ($M_w$ ~ 2900), cannot contribute to construct long one-dimensional skeletons, i.e., aerogels with high bendability cannot be obtained. The HLB value probably has a little influence on the determination of the pore skeleton structure. This is because P105 and L64 have led to different skeletal ratios ($3.8 \pm 0.5$ and $0.95 \pm 0.3$, respectively), though they have similar HLB values in 12–18. It is known that one-dimensional self-assembly of silica nanoparticles occurs in the presence of Pluronic surfactants due to the difference of steric crowding by hydrogen-bonded surfactant molecules on the particles[51,52]. In the present case, it can be deduced that polycondensation proceeds preferentially in 1D direction in a similar manner under an additional effect of asymmetric viscoelastic properties of gelling phase and solvent[53]. Such network

microstructures as the result of viscoelastic phase separation were shown to be advantageous in obtaining materials with low density and high mechanical strength[54]. In terms of transparency, a moderately low HLB value (e.g., P94: 13.5 and L64: 12–18) may help to make PMSQ aerogels more transparent presumably because of the effective suppression of phase separation, while a high HLB value (e.g., F127: 18–23 and F68: > 24) may not contribute to further improvement of the transparency. As shown in the previous study, Pluronic surfactants with too low HLB values (e.g., P123: 7–12) cannot make PMSQ aerogels transparent due to low ability for suppression of phase separation[24].

In light of the present results, it is reasonable to use a PEO-*b*-PPO-*b*-PEO-type surfactant with as high a molecular weight as possible and a moderately small HLB value (somewhere between 12–18) in order to optimize the transparency and bendability of PMSQ aerogels. In this study, we obtained the PMSQ aerogel with high transparency ($T_{550} = 90\%$) and the highest bendability ($\varepsilon_{max,20} = 75\%$ and $\varepsilon_{max,60} = 19\%$) by using Pluronic P105 ($M_w$ ~ 6500 and HLB value: 12–18). In addition, we obtained a PMSQ aerogel with the highest transparency ($T_{550} = 96\%$) by using Pluronic P94 ($M_w$ ~ 5000 and HLB value: 13.5). This aerogel (PMSQ-P94) has lower bendability ($\varepsilon_{max,60} = 10\%$) than that of PMSQ-P105 probably due to the even finer pore structure. However, it should be attributed to the one-dimensionally long pore skeletons (high skeletal ratio), that the bendability of PMSQ-P94 is slightly better than that of a previous aerogel (PMSQ-prev: $\varepsilon_{max,60} = 9\%$). Although it would be possible to extend the discussion by investigating aerogels prepared using Pluronic with low molecular weight and high HLB value or with high molecular weight and low HLB value, such surfactants are not available.

Here we stress again that this is the first report realizing a high-level combination of visible transparency and mechanical flexibility. In previous literature, silica aerogels derived from tetramethoxysilane (TMOS) and those from partially hydrolyzed tetraethoxysilane (TEOS) were reported to show comparably high transmittance[3,47], but no improvement of the mechanical properties are reported. Improvement of mechanical flexibility has been demonstrated in polymer-reinforced aerogels and other aerogels consisting of specific nanostructures (e.g., CNT aerogels and graphene aerogels); however, their visible transparency remains low (or zero)[7,8,15]. Transparent aerogels were prepared with TEMPO-oxidized cellulose nanofibers[13] and chitosan nanofibers[14], while these are not mechanically resilient because of the weak cross-links and low cross-linking density between the fibers. In the present study, however, we report the highly transparent PMSQ aerogels constructed with fiber-like structure, which are highly flexible and resilient in compression and especially in bending. In this study, there is still a trade-off relation between the transparency and the bendability, especially in the samples with very high transparency. In future studies, there will be a high possibility to realize both mechanical flexibility and transparency at an even higher level by optimizing the molecular structure of surfactant and the solid constituent of aerogels.

In summary, we report a synthetic strategy to design highly transparent aerogels with high bending flexibility. Taking PMSQ aerogel as an example, we successfully prepared highly transparent and highly bendable low-density resilient PMSQ aerogels by the simple combination of using PEO-*b*-PPO-*b*-PEO-type surfactants (Pluronic) as phase separation suppressors and structure determining agents and tetramethylammonium hydroxide (TMAOH) as a polycondensation catalyst in aqueous systems. All the resulting PMSQ aerogels are highly transparent ($T_{550} = 83-96\%$), and some of them have one-dimensionally long pore skeletons (mesoscopic fiber-like structure with high skeletal ratio), which would contribute significantly to the improved bendability. The mesoscale structure is controlled by changing Pluronic: the higher molecular weight can result in more fiber-like structure with higher skeletal ratio, whereas the lower molecular weight can result in more particle aggregation-like structure

with lower skeletal ratio; a moderate HLB value (somewhere between 12–18 for PMSQ) may result in the finest pore structure, the highest transparency. The most bendable PMSQ aerogel obtained in this study has at least about twice better bendability than the previous aerogels with superior bendability. The most transparent PMSQ aerogel obtained in this study is even tough with fine fiber-like structure and one of the most transparent aerogels ever reported with low haze, which shows highly clear appearance almost comparable to glass. This study provides an opportunity to realize highly insulating flexible materials with high transparency through the control of the mesoscopic skeletal features and structural size.

## Methods

### Chemicals and materials

Methyltrimethoxysilane (MTMS) was purchased from Shin-Etsu Chemical Co., Ltd. (Japan). Acetic acid (HOAc, ≥99.7%), distilled water ($H_2O$), methanol (MeOH, ≥99.5%), 2-propanol (IPA, ≥99.0%), and *N,N*-dimethylformamide (DMF, ≥99.5%) were purchased from Kishida Chemical Co., Ltd. (Japan). Aqueous tetramethylammonium hydroxide (TMAOH, *ca.* 25%) and cationic surfactant *n*-hexadecyl-trimethylammonium chloride (CTAC) were purchased from Tokyo Chemical Ind. Co., Ltd. (Japan). Urea was purchased from Hayashi Pure Chemical Ind., Ltd. (Japan). Nonionic surfactant Pluronic F127 ($EO_{106}PO_{70}EO_{106}$, $M_w$ ~ 12,600) and Pluronic L64 ($EO_{13}PO_{30}EO_{13}$, $M_n$ ~ 2900) were purchased from Sigma-Aldrich Co., LLC (USA). Nonionic surfactant Synperonic P105 ($EO_{37}PO_{56}EO_{37}$, $M_w$ ~ 6500) was kindly donated by Croda Japan K.K. Nonionic surfactant Pluronic P94 ($EO_{26}PO_{48}EO_{26}$, $M_w$ ~ 5000) was purchased from BASF (Germany). Nonionic surfactant Pluronic F68 ($EO_{76}PO_{29}EO_{76}$, $M_w$ ~ 8400) was kindly donated by ADEKA Co., Ltd. (Japan). All reagents were used as received.

### Materials synthesis

**PMSQ aerogels prepared by using nonionic surfactant and TMAOH.** The starting compositions are listed in Supplementary Table 2. In a glass tube, 5.0 mL of 5 mM HOAc and 5.0 mL of MTMS were mixed and continuously stirred for 15 min at room temperature for hydrolysis of MTMS. A given amount of surfactant and $H_2O$ were subsequently added to the obtained homogeneous sol (in the case of PMSQ-P105DMF, a given amount of DMF was added instead of $H_2O$) and the solution was kept stirred until it became homogeneous (typically, for ~1 h) at room temperature. Then, the resulting sol was cooled in the reaction vessel in an ice bath for 30 min. Under moderate stirring to avoid a bubble formation in the ice bath, a given amount of TMAOH aq. was carefully added, and after stirring for 3 min, the mixture was transferred to an airtight container and allowed to gel and age at room temperature for 1 h. Gelation occurred in ~15 min. The obtained gel was kept at 60 °C for 3 days for further aging. The aged gel was soaked in solvents at 60 °C for at least 8 h for washing and solvent exchange. In order to prevent the gel from cracking due to osmotic pressure, the composition of the solvent was changed in the following order: $H_2O$:MeOH = 100:0, 90:10, 70:30, 50:50, 30:70, 10:90, 0:100 in volume ratio, and finally exchanged to IPA. In the case of the last two steps in pure MeOH and IPA, the solvent exchanging processes were repeated 5 times and 3 times, respectively. The obtained alcogel was supercritically dried at 80 °C under 14 MPa for 10 h with carbon dioxide.

**PMSQ aerogels prepared by following our previous research for testing the hypothesis of phase separation phenomenon[20] and for comparison with this work[21,25].** In a glass tube, 6.0 mL of 5 mM HOAc, 0.50 g of urea, and a given amount of Pluronic F127 or CTAC were mixed and continuously stirred until the mixture became homogeneous (typically, for 30 min). Then, 5.0 mL of MTMS was added and the mixture was kept stirred at room temperature for 30 min to allow hydrolysis of MTMS. In another case for preparing the PMSQ aerogel

with colloidal aggregate structure (PMSQ-prev), 12 mL of 5 mM HOAc, 3.0 g of urea, 0.40 g of CTAC, and 5.0 mL of MTMS were mixed in a similar way. The obtained homogeneous mixture was kept at 60 °C for 4 d for gelation and aging. The obtained gels were subjected to washing by solvent exchange using MeOH (for ≥8 h, each for 3–5 times) and IPA (for ≥8 h, each for 3 times). The resultant gels were then supercritically dried at 80 °C under 14 MPa for 10 h with carbon dioxide.

**Physical characterizations.** Bulk density ($\rho_b$) of the obtained aerogels was determined by measuring the diameter, height, and weight of cylindrical samples. The spectral data of transmittance of the aerogels were obtained with a V-670 UV-vis-NIR spectrophotometer (JASCO Co., Japan) equipped with an integrated sphere. The value of total transmittance at 550 nm wavelength was converted to the one in a 10 mm-thick sample according to the Lambert-Beer equation and denoted as total transmittance ($T_{550}$). The pore/skeletal structures of aerogels were observed under a field-emission scanning electron microscopy (FE-SEM: Regulus 8220, Hitachi High-Tech Co., Japan) and a scanning transmission electron microscopy (STEM: JEM-1400Plus, JEOL Co., Japan). Before observation, aerogels were gently crushed with an ultra-fine sandpaper and the crushed aerogels on the sandpaper were scattered on a carbon tape attached to a sample stage (FESEM) or a grid (STEM). All structural observations were performed without any conductive coatings to avoid influence on the microstructure. Pore structures of the samples were characterized from nitrogen isotherms at 77 K with BELSORP-max (MicrotracBEL Corp., Japan). The samples were degassed at 80 °C overnight prior to each measurement. Specific surface area was calculated by the Brunauer-Emmett-Teller (BET) method and pore size distribution was by the Barrett-Joyner-Halenda (BJH) method from the adsorption branch.

The chemical structures were investigated by $^{29}$Si and $^{13}$C CP/MAS NMR experiments using an NMR spectrometer equipped with a double-resonance 4-mm MAS probe (Bruker Avance III 800US Plus, Germany, operated at 158.96 MHz, and 201.20 MHz for $^{29}$Si, and $^{13}$C, respectively). The measurements were carried out at room temperature at the MAS frequency of 12 kHz throughout the study. Hexamethylcyclotrisiloxane and hexamethylbenzene were used as external reference materials for $^{29}$Si, and $^{13}$C, respectively. Cross-polarization contact times were 5.5 ms for $^{29}$Si and 4.5 ms for $^{13}$C. The $^{29}$Si and $^{13}$C field strengths of 50 and 100 kHz were used during the CP period. A total of 8192 scans was accumulated for the $^{29}$Si measurements. The thermal properties of the samples were investigated by thermogravimetry-differential thermal analysis (TG-DTA: TG-DTA 8122, Rigaku Co., Japan). The samples were heated with a rate of 5 °C min$^{-1}$ in air supplied at *ca.* 100 mL min$^{-1}$. The mechanical properties of the aerogels were investigated by a material tester (AUTOGRAPH AG-X plus, Shimadzu Co., Japan). For three-point bending tests, cylindrical samples with diameter of about 10 mm and length of about 80 mm were used. The span length was fixed at 20 mm or 60 mm, and the cross-head speed was fixed at 0.5 mm min$^{-1}$.

The stress ($\sigma$) and bending strain ($\varepsilon$) were calculated from the following equations, in which $D$ is the diameter of a cylindrical sample, $L$ is span length, $\Delta l$ is displacement length, and $F$ is loaded force.

$$\sigma = \frac{8L}{\pi D^3} \cdot F$$

$$\varepsilon = \frac{6D}{L^2} \cdot \triangle l$$

For uniaxial compression–decompression tests, cylindrical specimens with diameter of about 10 mm and height of about 10 mm were used. The specimens were prepared by cutting cylindrical samples with a custom-made electric saw. The cross-head speed was fixed at

0.5 mm min$^{-1}$. The thermal conductivity was determined by a heat flow meter (HFM 436 Lambda, Netzsch, Germany) at the mean temperature of 25 °C with a plate sample of 100 × 100 × 8 mm.

## Data availability
The data that support the findings of this study have been included in the main text and Supplementary Information. All other relevant data supporting the findings of this study are available from the corresponding authors upon request.

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

## Acknowledgements

This work was financially supported by collaborative research with tiem factory, Inc. (Japan), under the development phase of practical application, strategic energy saving (SES) and technical innovation program (P12004), New Energy and Industrial Technology Development Organization (NEDO), Japan (K.K.).

## Author contributions

R.U., K.K. and Y.H. designed the project. R.U. performed the experiments and wrote the paper. Y.H. prepared the manuscript with input from all authors. A.M. and H.K. contributed to the NMR measurement. K.K. and K.N. provided the research facility of materials synthesis and characterization. K.K. supervised the project and contributed to writing the paper. All authors reviewed and commented on the manuscript.

## Competing interests

The authors declare no competing interests.
