## [Peer review file · Nature Communications]

REVIEWER COMMENTS

Reviewer #1 (Remarks to the Author):

This paper reported the preparation of aerogels with both transparency and bending flexibility based on PMSQ aerogels with various types of Pluronics as surfactant to optimize the mesoscale structure. The authors have conducted some analysis on this system and obtained relevant data. However, the reviewer's primary concern is the missing novelty of the submitted contribution compared with the already existing similar aerogels in the literature. This manuscript does not, in my opinion, contain any significant new contribution and thus, it is not acceptable for consideration by the journal of Nature Communications based on the following reasons:

(1) The major concern which needs to be addressed is the novelty of the work. The main challenge seems to be designing structures (in this case silica-based aerogels) with both high flexibility and transparency. However, a quick search in the literature (from the authors' previous works and others) show that such aerogels exist and have been synthesized (look at references: Chem. Mater. 2020, 32, 4, 1595–1604 , Chem. Mater. 2018, 30, 8, 2759–2770 , ACS Nano 2018, 12, 1, 521–532 , and Journal of Colloid and Interface Science 2022, Pages 1101-1110). Also, PMSQ-based aerogels have been frequently reported with overall excellent properties (look at references: Journal of Sol-Gel Science and Technology volume 89, pages166–175 (2019), Powder Technology Volume 373, August 2020, Pages 716-726).

(2) Another aspect which makes the novelty ambiguous and weakens the quality of this work is using the same surfactant introduced previously in other works. Pluronic has been used in many works as a surfactant in synthesis of aerogels (even PMSQ aerogels; look at references: Journal of Colloid and Interface Science 2011, Pages 336-344 , J. Mater. Chem. A, 2014, 2, 6525-6531). Moreover, other types of surfactants have been used in similar cases and the discussions need to reflect the major differences and improvements achieved with respect to those works (Macromolecular communications, Volume 44, Issue2, January 2023, 2200628 , Journal of Colloid and Interface Science Volume 485, 1 January 2017, Pages 152-158 , Microporous and Mesoporous Materials Volume 158, 1 August 2012, Pages 247-252).

(3) The use of surfactant itself needs to be justified. In literature it is stated that large monoliths need careful and extended washing of surfactant, because the residual surfactant causes serious shrinkage and cracks in the monoliths during drying. For this reason, a surfactant-free synthesis process was claimed to be desirable. This makes the use of surfactant questionable and requires clearly discussing the shortcomings of this work as well.

Reviewer #2 (Remarks to the Author):

This manuscript reports a new synthetic strategy to realize highly transparent aerogels with outstanding flexibility through the optimization of the mesoscale porous structure by surfactant-induced mesoscopic fiber assembly. The most significant finding of this work is that the structures can be modulated by using different surfactants, giving rise to different optical transmittance and mechanical flexibility of the resulting aerogels. However, it lacks a quantitative structure-property relationship, which is essential for guiding the design of transparent aerogels. Therefore, I cannot recommend the publication of this manuscript in the current form. However, if the following comments can be well addressed, I am willing to reconsider its suitability for publication in Nature Communications.

1. Figure 2: Two span lengths were used in the three-point bending tests, which yielded completed different mechanical properties of PMSQ-F127 in comparison to the other two. I am not sure what the

reason is for using two span lengths which can lead to confusing results. It is suggested that authors should adhere to relevant standard testing methods for the span of three-point bending test to generate meaningful results.

2. The authors attribute the different mechanical properties to the different porous structures, namely, particle aggregated structure vs. fiber-like structure. However, it is unclear how these two structures are defined. Why PMSQ-prev and PMSQ-P105DMF are categorized as particle aggregated structure while the other two fiber-like structures? Please provide quantitative criteria for the classification.

3. Figure 3b: the spectra of four aerogels look almost identical, which makes no sense to show them here. I understand that the authors wish to calculate the condensation degree from the peak area, but the difference is also marginal. How could a correlation between condensation degree and mechanical properties be established from such characterization?

4. Figure 3c: the comparison here does not give a clear conclusion regarding the correlation between mesoscale porous structure and the mechanical properties. For example, what is the reason for a higher modulus and strength of PMSQ-F105 than PMSQ-127 given they are both fiber-like structures? In addition, what causes a higher modulus but a lower strength and failure strain of PMSQ-P105DMF than PMSQ-P105? It would be more informative if quantitative relationships between mesoscale porous structure and mechanical properties can be established.

5. Figure 4b: Different surfactants lead to different transmittance values. What is the main structural factor determining the optical transmittance of different surfactants, HLB value or the molecular weight? It is important to establish a quantitative relationship to guide the design of transparent aerogel with different surfactants.

6. The authors use three-point bending tests to measure the flexibility of the aerogels. However, the three-point bending test does not provide elasticity of the aerogel. What about the elasticity under compression?

7. The properties, including transmittance, mechanical properties, and the thermal conductivity of the aerogels obtained in this work should be compared to existing materials in the literature to highlight the advantages of current aerogel.

8. Although many properties are characterized in the manuscript, it is not clear the target application of the transparent aerogel. If it is targeted to replace glass or act as an inner layer in the window, the reduced interior temperatures should be demonstrated in an outdoor test to show the practical usefulness of the current aerogel.

Reviewer #3 (Remarks to the Author):

Review on Nature Communications paper by Kanamori and co-workers.

This paper describes a systematic study on how the synthesis conditions (base type and concentration, surfactant type and concentration) can be tailored to improve excellent transparency and bendability. The microstructure is also investigated in detail and its effects on properties is discerned. The combination of properties really are beyond the current state of the art, the synthesis and characterization have been carried out well and described fully, and the manuscript is well written. There are some things that can be improved (none unsurmountable), hence, I see good publication prospects and recommend the following revisions to the manuscript.

1. NMR: changes in T2 concentration/peak intensity are very small and the data is not convincing that the observed changes are actually real and significant. Phasing of spectra may induce changes

(PMSQ P105 spectrum has a higher tail at around -73 ppm, which may be due to incorrect phasing. Also, CP can definitely distort intensities when we are discussing such small changes. I agree that this problem will not be huge for PMSQ, but at such high claimed accuracy, all bets are off.

2. Around line 250: bending and other mechanical properties are not expected to directly correlate with chemical structure, rather with microstructure (which may itself depend on chemical structure). I think the authors agree with this, but the phrasing is a bit weird. In each case, the chemical differences are insignificant in my opinion (unless the authors can rigorously show they are significant).

3. Porous structure: this is one of the key aspects of the paper: please show SEM and TEM images also in the main text, not only the SI. Define skeletal ratio also in main text.

4. Scheme 1 seems to imply low skeletal ratio = high density and high skeletal ratio = low density, whereas in reality all densities are basically the same. Please adjust schematic illustration and/or discuss density in this context.

5. The high transparency is one key selling point: please also show photographs of aerogels on a black background and as a "window", i.e. not touching the background but in front of some sort of scenery or person. This gives a better impression of the real-world optical quality.

6. Data presentation is not always friendly to the color-blind.

7. Much is discussed on skeletal ratio versus bendability etc., but no correlations are shown and data are not tabulated close together. Please add e.g. skeletal ratios to Figure 4d (and also add sample names there!)

8. I understand that 3 point bending is the main improvement here, but please also show compression data (including compression, decompression cycles), as such data is more common and provides an easy comparison of performance to other studies.

9. Also for 3 point bending, it would be good to have some strain recovery data.

Reviewer #1 (Remarks to the Author):

This paper reported the preparation of aerogels with both transparency and bending flexibility based on PMSQ aerogels with various types of Pluronics as surfactant to optimize the mesoscale structure. The authors have conducted some analysis on this system and obtained relevant data. However, the reviewer's primary concern is the missing novelty of the submitted contribution compared with the already existing similar aerogels in the literature. This manuscript does not, in my opinion, contain any significant new contribution and thus, it is not acceptable for consideration by the journal of Nature Communications based on the following reasons:

Reply: We appreciate your precious time to review our manuscript and important comments. We revised the manuscript in full accordance with your comments, so that the novelty of this work can be clearly understood. Now we believe the manuscript has been fully improved to solve your concerns and is suitable for publication in Nat. Commun.

In the revised manuscript, all the revised parts according to the reviewers' comments are highlighted in yellow, and those revised or moved to make the logical flow clearer are highlighted in green, with all the modifications recorded by the track change function of MS Word. The point-by-point replies are described as follows.

(1) The major concern which needs to be addressed is the novelty of the work. The main challenge seems to be designing structures (in this case silica-based aerogels) with both high flexibility and transparency. However, a quick search in the literature (from the authors' previous works and others) show that such aerogels exist and have been synthesized (look at references: Chem. Mater. 2020, 32, 4, 1595–1604, Chem. Mater. 2018, 30, 8, 2759–2770, ACS Nano 2018, 12, 1, 521–532, and Journal of Colloid and Interface Science 2022, Pages 1101-1110). Also, PMSQ-based aerogels have been frequently reported with overall excellent properties (look at references: Journal of Sol-Gel Science and Technology volume 89, pages166–175 (2019), Powder Technology Volume 373, August 2020, Pages 716-726).

Reply: We appreciate your important comment on the novelty of the present manuscript. As you pointed out, there are many related works on mechanically flexible aerogels. Here please allow us to emphasize that “flexible and transparent” aerogels in “monolithic form” have not been reported before, which is the major novelty of the present work. It is widely accepted that mechanical flexibility is crucial for extended application of aerogels in addition to the fact that it is of fundamental importance in scientific research. Compression flexibility is of course important but bending flexibility is by far difficult to be achieved because bending contains tensile deformation, which stretches the pore skeletons apart. The handling ability of aerogels cannot be improved only by compression flexibility. Visible-light transparency is one of the most important features of aerogels that differentiate from other porous materials. Moreover, it should be emphasized other important properties such as low density and low thermal conductivity are maintained in the aerogels reported in the present manuscript.

These two properties, mechanical flexibility (in particular, bending flexibility) and high optical

transparency, have been realized at the same time in different material formats such as films, but never been realized in monolithic forms. Further, aerogels with EITHER mechanical flexibility OR high optical transparency have been reported extensively, e.g., in the papers you mentioned in the comment, as follows;

- Chem. Mater. 2020, 32, 4, 1595–1604: One of our previous papers. Mechanically flexible, but semi-transparent and bendable only in a thin membrane format.
- Chem. Mater. 2018, 30, 8, 2759–2770: One of our previous papers. Mechanically flexible after crosslinking the vinyl groups, but semi-transparent. Bendability is not reported.
- ACS Nano 2018, 12, 1, 521–532: One of our previous papers. Mechanically flexible, but semi-transparent and bendable only in a thin membrane format.
- Journal of Colloid and Interface Science 2022, Pages 1101-1110: Semi-transparent aerogels and bendability is not reported.
- Journal of Sol-Gel Science and Technology volume 89, pages166–175 (2019): One of our previous papers, in which our previous works are reviewed. No aerogels with both bending flexibility and high optical transparency are shown.
- Powder Technology Volume 373, August 2020, Pages 716-726: Mechanically flexible only against compression and optically opaque.

The novelty of this work, mechanically flexible, optically highly transparent in the monolithic format, has been accomplished by a rather simple modification of our reported process from methyltrimethoxysilane (MTMS). This in turn means that a simple improvement in the reaction system can lead to a drastic change in the aerogel properties. In many cases, reinforcement materials and/or processes such as polymer crosslinking and extended aging are applied to improve the mechanical properties. However, the process reported in the present manuscript does not rely on any additional materials and processes but simply rely on the morphological controls under surfactant-assisted, optimized basic condition. This strategy may be applicable to other chemical composition systems such as silica or polymer aerogels. This point is further explained in the following comment and reply.

In the revised manuscript, we added Table 1 to compare with other reported materials and emphasize the novelty of the present work. Additional texts in the corresponding parts have also been included in the revised manuscript in page 10 as: “As will be discussed in more detail later, the PMSQ aerogel with comparable high transparency and low bulk density obtained in the previous work²⁵ (PMSQ-prev) has an $\epsilon_{\max,20}$ of 30%, indicating that PMSQ-F127 and -P105 can be bent even higher. Table 1 shows comparisons with other aerogels that have shown comparable three-point bending behaviors to this work. Note that it is difficult to make completely accurate comparison with other aerogels obtained in previous studies because the results of three-point bending tests are strongly influenced by the testing condition and sample geometry. However, focusing only on bending flexibility, there is no aerogel that has as high bending flexibility as PMSQ-F127 or -P105. In addition, only silica aerogels have comparable high transparency to PMSQ-F127 or -P105 and other non-silica aerogels have significantly lower transparency. Therefore, it is reasonable to regard PMSQ-F127 and -P105 have an unprecedented combination of high transparency and high bending flexibility.”

(2) Another aspect which makes the novelty ambiguous and weakens the quality of this work is using the same surfactant introduced previously in other works. Pluronic has been used in many works as a surfactant in synthesis of aerogels (even PMSQ aerogels; look at references: Journal of Colloid and Interface Science 2011, Pages 336-344 , J. Mater. Chem. A, 2014, 2, 6525-6531). Moreover, other types of surfactants have been used in similar cases and the discussions need to reflect the major differences and improvements achieved with respect to those works (Macromolecular communications, Volume 44, Issue2, January 2023, 2200628 , Journal of Colloid and Interface Science Volume 485, 1 January 2017, Pages 152-158 , Microporous and Mesoporous Materials Volume 158, 1 August 2012, Pages 247-252).

Reply: As you commented, the use of the Pluronic surfactant has already been reported in the PMSQ system in our previous works. Surfactant is needed in the preparation of PMSQ aerogels to suppress phase separation of the hydrophobic PMSQ condensates in aqueous solution. While it is difficult to clarify the role of surfactant, in one of our previous works (RSC Adv., 2012, 2, 7166–7173), we deduced from NMR results that hydrophobic interaction between propylene oxide units in Pluronic surfactant and methyl groups in PMSQ is dominant to make the hydrophobic surface of PMSQ hydrophilic, so that phase separation is suppressed. Other surfactants have also been tried and it has been experimentally found that cationic surfactants like CTAB/CTAC are effective for suppressing phase separation, presumably by the similar hydrophobic interaction between alkyl chains in surfactant and methyl groups in PMSQ.

In the present work, we extended the kind of Pluronic surfactant and found surfactants with moderate HLB and molecular weight (P94 and L64) are more effective to suppress phase separation, while those with higher molecular weight (F127 and P105) leads to higher mechanical flexibility. Moreover, it was also found that the pH control by the strong base as condensation catalyst (tetramethylammonium hydroxide, TMAOH) allows homogeneous network formations, resulting in higher optical transparency. It must be emphasized that a simple combination of these two elements, Pluronic surfactant and strong base TMAOH, gave an outstanding impact in concomitant optical transparency and mechanical flexibility, which are crucial for aerogel applications with glasslike transparency, easier production and easier handling. Differences from other surfactants, e.g., CTAC, are discussed in the manuscript; aerogels prepared in the presence of CTAC consist of spherical colloid-like structure so the mechanical flexibility is not as high as those prepared with Pluronics. Moreover, the chemical structure (the lengths of ethylene oxide and propylene oxide units) of Pluronic-type surfactant is controllable. If the chemical structure is optimized to allow extended combinations of HLB and molecular weight in future works, then it would be possible to design aerogels with even better physical properties. For these reasons we believe the use of already reported surfactant does not decrease the novelty nor weaken the quality of this manuscript.

Differences from the papers you listed are as follows;

- Journal of Colloid and Interface Science 2011, Pages 336–344: One of our previous papers, mainly discussing the pore structure controls over micrometer scale in PMSQ system prepared with F127. Urea was used as the base source. Highly transparent and mechanically flexible aerogel is not reported.
- J. Mater. Chem. A, 2014, 2, 6525–6531: Thermal conductivity of the aerogels reported in the above

paper is discussed. Low transparency and only compression flexibility is reported.

- Macromolecular communications, Volume 44, Issue2, January 2023, 2200628: PMSQ-cellulose nanofibers (CNF) aerogels prepared with CTAB or F127, based on our original research. Urea was used as the base source. Low-transparent aerogels are reported with mechanical compressibility, but no recovery from compression are reported.
- Journal of Colloid and Interface Science Volume 485, 1 January 2017, Pages 152–158: PMSQ aerogels prepared with CTAB based on our original research. No transparent, flexible aerogels are reported.
- Microporous and Mesoporous Materials Volume 158, 1 August 2012, Pages 247–252: One of our previous papers. Relationship between CTAC surfactant and physical properties of aerogels are reported, but no mechanical flexibility is reported.

(3) The use of surfactant itself needs to be justified. In literature it is stated that large monoliths need careful and extended washing of surfactant, because the residual surfactant causes serious shrinkage and cracks in the monoliths during drying. For this reason, a surfactant-free synthesis process was claimed to be desirable. This makes the use of surfactant questionable and requires clearly discussing the shortcomings of this work as well.

Reply: As you commented, surfactant may become problematic in the industrial production of the PMSQ aerogels. However, as replied in the above (2), surfactant is still needed to suppress phase separation of the hydrophobic PMSQ in aqueous solution. The point here, however, is to fabricate desirable porous structures in order to maximize the mechanical flexibility, which is the worst drawback of aerogels, while keeping (or even improving) the characteristic optical transparency. In our fundamental research described in this manuscript, we have succeeded to establish an excellent combination of mechanical flexibility and optical transparency by using Pluronic surfactant with adequate characters (HLB and molecular weight) and strong base TMAOH. Correlations between the kind and character of Pluronic surfactant, molecular structure of the network, porous structure, and their physical properties have been systematically investigated in this manuscript. We found the fiber-like pore skeletons are formed in this condition and believe this is one crucial step for improving the applicability of aerogels. We already know that the use of simple organic solvent, instead of surfactant, can form transparent aerogels in PMSQ (not reported), but mechanical flexibility has not been realized without the use of surfactant. However, as you commented, it will also be important to avoid the use of surfactant in future works, while maintaining the mechanical flexibility and optical transparency reported in this manuscript. We are also trying to prepare transparent, monolithic PMSQ gels from a simple aqueous system by suppressing phase separation in a different mechanism, which we hope will be presented in our future papers.

Reviewer #2 (Remarks to the Author):

This manuscript reports a new synthetic strategy to realize highly transparent aerogels with outstanding flexibility through the optimization of the mesoscale porous structure by surfactant-induced mesoscopic fiber assembly. The most significant finding of this work is that the structures can be modulated by using different surfactants, giving rise to different optical transmittance and mechanical flexibility of the resulting aerogels. However, it lacks a quantitative structure-property relationship, which is essential for guiding the design of transparent aerogels. Therefore, I cannot recommend the publication of this manuscript in the current form. However, if the following comments can be well addressed, I am willing to reconsider its suitability for publication in Nature Communications.

Reply: We appreciate your precious time to review our manuscript and important comments to improve the manuscript. We revised the manuscript in full accordance with your comments. Now we believe the manuscript has been fully improved to meet your criteria and is suitable for publication in Nat. Commun.

In the revised manuscript, all the revised parts according to the reviewers' comments are highlighted in yellow, and those revised or moved to make the logical flow clearer are highlighted in green, with all the modifications recorded by the track change function of MS Word. The point-by-point replies are described as follows.

1. Figure 2: Two span lengths were used in the three-point bending tests, which yielded completed different mechanical properties of PMSQ-F127 in comparison to the other two. I am not sure what the reason is for using two span lengths which can lead to confusing results. It is suggested that authors should adhere to relevant standard testing methods for the span of three-point bending test to generate meaningful results.

Reply: The reason for employing two different span lengths is to show results comparable with reports by other researchers (span length of 20 mm is commonly used for aerogel measurements) and to show more reasonable results with natural curvature at 60 mm. In fact, the relationship between mechanical properties and aspect ratio of the specimen has been discussed by Scherer et al. (Journal of Non-Crystalline Solids 354 (2008) 4556–4561), in which they reported meaningful results can be obtained with samples in rather high aspect ratios; in the case of three-point bending with the sample 0.18 g cm^{-3} , the ratio of span length (s) and sample diameter (D) should be $s/D > 6$. There is no standardized method for mechanical testing of aerogels, so we presented here the results from two different measurement conditions to make the results comparable with others and to obtain meaningful results.

However, we agree that our presentation of the data is a little confusing, so we have revised this point by deleting the result of 60 mm span from Fig. 2b (old) and use only the results of 20 mm span for comparative studies in Fig. 3b (new). After such discussions, the results of 60 mm span are shown in Fig. 5c (new) to ensure the more reasonable discussion of mechanical flexibility. Additional texts in the corresponding parts have also been included in page 10 in the revised manuscript as: "It is worth noting that the bending test with a short span does not exhibit natural curvature in thick specimens; i.e., an accurate evaluation of bending performance has not been done³⁶. We therefore have also performed three-

point bending tests with a long span length (60 mm). In this testing condition, PMSQ-F127 shows large bending deformation (Fig. 2c), which is almost the same as PMSQ-P105. The obtained stress–strain curves are shown later in Fig. 4c.”

2. The authors attribute the different mechanical properties to the different porous structures, namely, particle aggregated structure vs. fiber-like structure. However, it is unclear how these two structures are defined. Why PMSQ-prev and PMSQ-P105DMF are categorized as particle aggregated structure while the other two fiber-like structures? Please provide quantitative criteria for the classification.

Reply: The semi-quantitative characterizations of the porous structure are needed to clarify the structure-property relationship. We therefore extracted physical data from FESEM and TEM images. As commented, however, it is difficult to unambiguously define the porous structure from the structural data. In particular in the original manuscript, the two contrastive terms “fibrous skeletons” and “particle aggregated structure” were used, but we agree that the structures cannot be defined as one by one. We therefore revised such expressions to high (low) skeletal ratio structure, with supportive uses of the terms “fiber-like” (“particle aggregation-like”). In addition, we redefined the skeletal ratio parameter by (length of the skeleton)/(thickness of the skeleton). We believe this definition, in aspect ratio of the fiber-like parts, is more reasonable to evaluate the porous structure and make correlations with mechanical properties. Detailed definition is presented in Fig. 3c and the results are listed in Fig. 3d, 5d, and Supplementary Table 3. Additional texts in the corresponding parts have also been included in page 12 in the revised manuscript as; “To clarify differences in pore structure, we introduce the concept of skeletal ratio (Fig. 3c). The pore skeletons can be classified into (i) 1D skeletons and (ii) their connections (nodes). The average length and thickness of the branched 1D skeletons are denoted as skeleton length and skeleton thickness, respectively. The skeletal ratio is defined as (skeleton length)/(skeleton thickness). A high skeletal ratio gives the appearance of a fiber-like structure, while a low skeletal ratio gives the appearance of a typical particle aggregation-like structure (Fig. 3a). We calculated some kinds of size parameters, including skeletal ratio, from high-resolution FE-SEM and TEM images (see Supplementary Table 3 for details). PMSQ-F127 and -P105 have higher skeletal ratios (3.5 ± 0.6 and 3.8 ± 0.5 , respectively) than those of PMSQ-prev and -P105DMF (1.9 ± 0.2 and 1.7 ± 0.5 , respectively), whereas these four samples have similar bulk density and transparency (Fig. 3d).”

Figure 3. Control of the mesoscale structure of the PMSQ aerogel, and properties of the PMSQ aerogel samples with different mesoscale structures. (a) A schematic of controlling the mesoscale structure. (b) Stress–strain curves obtained from three-point bending of the PMSQ aerogel samples with 20 mm span. (c) A schematic of the pore skeletons with nodes. Skeletal ratio is defined as [skeleton length]/[skeleton thickness]. (d) Obtained properties of the PMSQ aerogel samples. [a] Bulk density. [b] Light transmittance at 550 nm through 10-mm equivalent specimen. [c] Calculated by using the means with a 95% confidence interval of [skeleton length] and [skeleton thickness]. The error shows the estimated maximum error. [d] Maximum strain at failure of the three-point bending of the PMSQ aerogel samples with 20 mm span.

3. Figure 3b: the spectra of four aerogels look almost identical, which makes no sense to show them here. I understand that the authors wish to calculate the condensation degree from the peak area, but the difference is also marginal. How could a correlation between condensation degree and mechanical properties be established from such characterization?

Reply: We agree with your comment here that the slight difference in the molecular scale (NMR) does not influence the mechanical properties much. Although we know from experience that even a small difference of crosslinking density (or residual silanols) can impact the spring-back behavior in PMSQ, we did not observe it in the present research. Rather, we should focus on the correlations between mesoscale porous structure and mechanical properties. To avoid confusions, we moved the NMR results to Supplementary Fig. 9, and made the discussion on this point a little simpler (page 13 in the revised manuscript). Please note that we performed the measurement again with a higher number of accumulation number (1024 → 8192 times per measurement, in page 25).

Revised as; “Supplementary Figure 9 shows the comparison of solid-state ²⁹Si cross-polarization magic angle spinning (CP/MAS) nuclear magnetic resonance (NMR) spectra of the PMSQ aerogels (-F127, -P105, -prev, and -P105DMF). Note that we have confirmed that there is no obvious difference between CP/MAS and single-pulse measurements in the PMSQ system. Peaks around -67 ppm and -57 ppm correspond to fully condensed (CH₃Si(OSi)₃, T₃) and doubly condensed (CH₃Si(OSi)₂(OH/CH₃), T₂) silicon species,

respectively. Although there are only negligible differences in the peak shape among these aerogels, the condensation degree values, $(T3+2/3T2)/(T3+T2)$, calculated from the peak areas, are slightly different. The condensation degree is 97.4%, 98.3%, 97.7%, and 97.6% for PMSQ-F127, -P105, -prev, and -P105DMF systems, respectively. In general, from the point of view of molecular-level structures, the lower condensation degree or cross-linking density should result in lower modulus and higher bendability. In the case of trifunctional MTMS, elasticity or resilience may be sacrificed due to the more residual alkoxy/hydroxy groups in the less cross-linked network. In the present case, however, there is no clear correlation between the bending flexibility and cross-linking density: PMSQ-P105 is the most bendable and has the highest cross-linking density, but the second most bendable aerogel, PMSQ-F127, has the lowest cross-linking density; also, both PMSQ-F127 and -prev have almost the same ^{29}Si CP/MAS NMR spectra and cross-linking density, but PMSQ-F127 can be bent much higher than PMSQ-prev.”

4. Figure 3c: the comparison here does not give a clear conclusion regarding the correlation between mesoscale porous structure and the mechanical properties. For example, what is the reason for a higher modulus and strength of PMSQ-F105 than PMSQ-127 given they are both fiber-like structures? In addition, what causes a higher modulus but a lower strength and failure strain of PMSQ-P105DMF than PMSQ-P105? It would be more informative if quantitative relationships between mesoscale porous structure and mechanical properties can be established.

Reply: We are sorry that the flow of discussion was not good so it was difficult to understand, while there is a fact that very “quantitative” understanding is difficult since many structural factors, such as molecular level structure and mesoscale porous structures (skeleton length, thickness, and their ratio, etc.), play important roles in the mechanical properties. However, here we can conclude that the mechanical properties observed in the present study can be elucidated from the factors: skeleton length, thickness, and their ratio. This is because all the aerogels have the same chemical composition (PMSQ), similar crosslinking density (from NMR), similar density and highly homogeneous porous structures, so only the difference is the skeletal parameters given in Supplementary Table 3. Overall, mechanical flexibility becomes higher with the higher skeletal ratio, and the modulus is largely influenced by the skeleton length and thickness.

When comparing PMSQ-F127 and PMSQ-P105, the higher modulus for the latter can be explained by the shorter skeletal length (higher number density of branching points or nodes), because the more rigid nodes contribute to the modulus compared to the less-rigid fiber-like skeleton parts. In PMSQ-P105DMF and PMSQ-P105, the major difference is the skeletal ratio (1.7 vs. 3.8, respectively). The lower skeletal ratio in PMSQ-P105DMF, meaning the aggregate-like structure, results in the higher modulus, and breaks at lower strain (Fig. 3b) because the number density of the nodes, in which the stress is concentrated when deformed, is higher. In order to make the discussion clearer, we added the summary part on the structure-mechanical property relationship in Discussion in pages 18–22 in the revised manuscript.

5. Figure 4b: Different surfactants lead to different transmittance values. What is the main structural factor

determining the optical transmittance of different surfactants, HLB value or the molecular weight? It is important to establish a quantitative relationship to guide the design of transparent aerogel with different surfactants.

Reply: Surfactant is needed in the preparation of PMSQ aerogels to suppress phase separation of the hydrophobic PMSQ condensates in aqueous solution. While it is difficult to clarify the role of surfactant, in one of our previous works (RSC Adv., 2012, 2, 7166–7173), we deduced from NMR results that hydrophobic interaction between PO units in Pluronic surfactant and methyl groups in PMSQ is dominant to make the hydrophobic surface of PMSQ hydrophilic, so that phase separation is suppressed. Other surfactants have also been tried and it has been experimentally found that cationic surfactants like CTAB/CTAC are effective for suppressing phase separation, presumably by the similar hydrophobic interaction between alkyl chains in surfactant and methyl groups in PMSQ.

In the present work, we extended the kind of Pluronic surfactant and found surfactants with moderate HLB values and lower molecular weight (P94 and L64) are more effective to suppress phase separation, which helps to maximize transparency, while those with higher HLB values and higher molecular weight (e.g., F127 and P105) leads to a little lower transparency.

Experimentally, Pluronic surfactants with middle HLB (~ 12) work well for obtaining more transparent PMSQ aerogels, and those with higher HLB (> 18) lead to slightly lower transparency. The summary on the relationships between surfactant and physical properties is added in the new section “Discussion” in the revised manuscript pages 18–22 for making conclusions clearer. However, we have to mention that the relationships we obtained here are not complete, because commercially available kinds of Pluronic surfactants are limited, e.g., Pluronic surfactants with high HLB values always have large molecular weight. Discussions by employing Pluronic surfactants with low HLB and high molecular weight; and those with high HLB and low molecular weight are lacking. We will clarify this point in our future works by using tailor synthesized Pluronic surfactants.

6. The authors use three-point bending tests to measure the flexibility of the aerogels. However, the three-point bending test does not provide elasticity of the aerogel. What about the elasticity under compression?

Reply: We are sorry we did not show the results of compression tests. We added the compression data in Supplementary Fig. 6 and 7, in which the prepared samples (data are shown for PMSQ-F127 and PMSQ-P105) are compressible to 50% without break and show spring-back to 80%. After relaxation at 110 deg C for two hours, the PMSQ-F127 showed the nearly perfect spring-back. Additional texts in the corresponding parts have also been included in page 11 in the revised manuscript as; “Similarly, residual strains were observed on specimens of PMSQ-F127 and -P105 after 50% uniaxial compression test (Supplementary Fig. 6). In this case, however, the specimens recover nearly to their original state by heating at 110 °C for a few hours (Supplementary Fig. 7). These results indicate that both PMSQ-F127 and -P105 are viscoelastic.”

Supplementary Figure 6. Stress–strain curves obtained by a uniaxial compression–decompression test with 50% strain on PMSQ-F127 and -P105. There are some residual strains on both PMSQ-F127 and -P105 after the test.

Supplementary Figure 7. Photographs of viscoelastic behavior of PMSQ-F127 during a uniaxial compression–decompression test with 50% strain. The specimen is cylindrical, ca. 10 mm diameter and height. The compressed specimen is approximately 80% of the pre-test height (from 10.3 to 8.60 mm). However, the compressed specimen recovers its original shape (10.0 mm height) by heating at 110 °C for 2 hours.

7. The properties, including transmittance, mechanical properties, and the thermal conductivity of the aerogels obtained in this work should be compared to existing materials in the literature to highlight the advantages of current aerogel.

Reply: We appreciate this comment regarding comparison with different aerogels. We added the comparisons in Table 1 in the revised manuscript, and show the present PMSQ aerogels are outstanding in terms of high-level combination of optical transparency and mechanical flexibility in low-density, monolithic samples. However, no thermal conductivity data are given in these reports on bendable aerogels. The thermal conductivity of 14.5 mW/(m K) measured on one of the present samples (PMSQ-P94) is in the same range as that of standard silica aerogels and PMSQ aerogels. We therefore believe the present materials are one of the lowest thermal conductivity aerogels ever reported. Additional texts in the corresponding parts have also been included in the revised manuscript.

Table 1. Comparison of the properties of the aerogels obtained in this work with reported aerogels.

Sample	ρ_b^*	Optical	ϵ_{max}^\ddagger	Reference
--------	------------	---------	---------------------------	-----------

	/g cm ⁻³	transparency†	/%	
PMSQ-F127	0.13	Transparent (83%)	51	This work
PMSQ-P105	0.12	Transparent (90%)	75	This work
Polymer-crosslinked silica	0.63	Opaque	40	[32]
Hexylene-bridged poly(silsesquioxane)	0.093	Opaque	40	[33]
Surface-modified hexylene-bridged poly(silsesquioxane)	0.22	Transparent (56%)	38	[34]
Silica	0.20	Not shown	8	[34]
POSS-based polysiloxane	0.20–0.21	Opaque	16–18	[35]

*Bulk density.

†Estimated from the appearance of the aerogels. The light transmittance values at 550 nm through 10-mm equivalent specimen are shown in the parentheses.

‡The maximum strain at failure estimated from stress–strain curves obtained from three-point bending of aerogels.

8. Although many properties are characterized in the manuscript, it is not clear the target application of the transparent aerogel. If it is targeted to replace glass or act as an inner layer in the window, the reduced interior temperatures should be demonstrated in an outdoor test to show the practical usefulness of the current aerogel.

Reply: We appreciate your comment on the possible applications of the present aerogels. As is studied extensively in many research groups, we also believe the present PMSQ aerogels are suitable for transparent insulation applications such as windows in houses and buildings. Here we report the thermal conductivity is low enough like 14.5 mW m⁻¹ K⁻¹ (PMSQ-P94) and these aerogels show glasslike transparency with low haze (1.7% at 5.34 mm in the case of PMSQ-P94) as shown in Figure 6. However, it is still difficult to prepare large-area (e.g., larger than 200 mm square) samples without cracks, so it is not possible to perform outdoor test at this moment. Instead, we added a photo taken outside to show the low-haze transparency of one of the samples (PMSQ-P94), which is shown in Supplementary Fig. 14.

Supplementary Figure 14. A photograph of PMSQ-P94 taken outdoor to show the potential as a window insulating material. The scenery behind the aerogel can be seen clearly because of the low haze of the sample.

Reviewer #3 (Remarks to the Author):

Review on Nature Communications paper by Kanamori and co-workers.

This paper describes a systematic study on how the synthesis conditions (base type and concentration, surfactant type and concentration) can be tailored to improve excellent transparency and bendability. The microstructure is also investigated in detail and its effects on properties is discerned. The combination of properties really are beyond the current state of the art, the synthesis and characterization have been carried out well and described fully, and the manuscript is well written. There are some things that can be improved (none unsurmountable), hence, I see good publication prospects and recommend the following revisions to the manuscript.

Reply: We appreciate your precious time to review our manuscript and high evaluation. We revised the manuscript in full accordance with your comments. Now we believe the manuscript has been fully improved and is suitable for publication in Nat. Commun.

In the revised manuscript, all the revised parts according to the reviewers' comments are highlighted in yellow, and those revised or moved to make the logical flow clearer are highlighted in green, with all the modifications recorded by the track change function of MS Word. The point-by-point replies are described as follows.

1. NMR: changes in T2 concentration/peak intensity are very small and the data is not convincing that the observed changes are actually real and significant. Phasing of spectra may induce changes (PMSQ P105 spectrum has a higher tail at around -73 ppm, which may be due to incorrect phasing. Also, CP can definitely distort intensities when we are discussing such small changes. I agree that this problem will not be huge for PMSQ, but at such high claimed accuracy, all bets are off.

Reply: We agree with your comment here that the slight difference in the molecular scale (NMR) does not influence the mechanical properties much. Although we know from experience that even a small difference of crosslinking density (residual silanols) can impact the spring-back behavior in PMSQ, we did not observe it in the present research. Rather, we should focus on the correlations between mesoscale porous structure and mechanical properties. To avoid confusions, we moved the NMR results to Supplementary Fig. 9, and made the discussion on this point a little simpler (page 13 in the manuscript) as follows. Please note that we performed the measurement again with a higher number of accumulation number and better phasing.

Revised as; "Supplementary Figure 9 shows the comparison of solid-state ^{29}Si cross-polarization magic angle spinning (CP/MAS) nuclear magnetic resonance (NMR) spectra of the PMSQ aerogels (-F127, -P105, -prev, and -P105DMF). Note that we have confirmed that there is no obvious difference between CP/MAS and single-pulse measurements in the PMSQ system. Peaks around -67 ppm and -57 ppm correspond to fully condensed ($\text{CH}_3\text{Si}(\text{OSi})_3$, T3) and doubly condensed ($\text{CH}_3\text{Si}(\text{OSi})_2(\text{OH}/\text{CH}_3)$, T2) silicon species, respectively. Although there are only negligible differences in the peak shape among these aerogels, the condensation degree values, $(T_3+2/3T_2)/(T_3+T_2)$, calculated from the peak areas, are slightly different. The

condensation degree is 97.4%, 98.3%, 97.7%, and 97.6% for PMSQ-F127, -P105, -prev, and -P105DMF systems, respectively. In general, from the point of view of molecular-level structures, the lower condensation degree or cross-linking density should result in lower modulus and higher bendability. In the case of trifunctional MTMS, elasticity or resilience may be sacrificed due to the more residual alkoxy/hydroxy groups in the less cross-linked network. In the present case, however, there is no clear correlation between the bending flexibility and cross-linking density: PMSQ-P105 is the most bendable and has the highest cross-linking density, but the second most bendable aerogel, PMSQ-F127, has the lowest cross-linking density; also, both PMSQ-F127 and -prev have almost the same ^{29}Si CP/MAS NMR spectra and cross-linking density, but PMSQ-F127 can be bent much higher than PMSQ-prev”

2. Around line 250: bending and other mechanical properties are not expected to directly correlate with chemical structure, rather with microstructure (which may itself depend on chemical structure). I think the authors agree with this, but the phrasing is a bit weird. In each case, the chemical differences are insignificant in my opinion (unless the authors can rigorously show they are significant).

Reply: We agree that the slight difference in the molecular structure in NMR does not influence the mechanical properties in the present case. Since the original writing was a bit confusing, we rewrote the corresponding part into; “In general, from the viewpoint of molecular-level structure, the lower condensation degree or cross-linking density would result in lower modulus and higher bendability. In the case of trifunctional MTMS, elasticity or resilience may be sacrificed due to the more residual alkoxy/hydroxy groups in the less cross-linked network. In the present case, however, there is no clear correlation between the bending flexibility and cross-linking density: PMSQ-P105 is the most bendable and has the highest cross-linking density, but the second most bendable aerogel, PMSQ-F127, has the lowest cross-linking density; also, both PMSQ-F127 and -prev have almost the same ^{29}Si CP/MAS NMR spectra and cross-linking density, but PMSQ-F127 can be bent much higher than PMSQ-prev” in page 13 of the revised manuscript (part of the revised paragraph in the above reply).

3. Porous structure: this is one of the key aspects of the paper: please show SEM and TEM images also in the main text, not only the SI. Define skeletal ratio also in main text.

Reply: In accordance with the reviewer’s comment, we showed SEM/TEM images in Fig. 1c-f, along with Supplementary Figs. 4,8,11. The definition of skeletal ratio is given clearly in Fig. 3c along with Supplementary Table 3. Please note that we redefined the skeletal ratio parameter by (length of the skeleton)/(thickness of the skeleton). We believe this definition, in aspect ratio of the fiber-like parts, is more reasonable to evaluate the porous structure and make correlations with mechanical properties.

Figure 3. Control of the mesoscale structure of the PMSQ aerogel, and properties of the PMSQ aerogel samples with different mesoscale structures. (a) A schematic of controlling the mesoscale structure. (b) Stress–strain curves obtained from three-point bending of the PMSQ aerogel samples with 20 mm span. (c) A schematic of the pore skeletons with nodes. Skeletal ratio is defined as [skeleton length]/[skeleton thickness]. (d) Obtained properties of the PMSQ aerogel samples. [a] Bulk density. [b] Light transmittance at 550 nm through 10-mm equivalent specimen. [c] Calculated by using the means with a 95% confidence interval of [skeleton length] and [skeleton thickness]. The error shows the estimated maximum error. [d] Maximum strain at failure of the three-point bending of the PMSQ aerogel samples with 20 mm span.

4. Scheme 1 seems to imply low skeletal ratio = high density and high skeletal ratio = low density, whereas in reality all densities are basically the same. Please adjust schematic illustration and/or discuss density in this context.

Reply: We appreciate this comment and agree that the schematic of the low skeletal ratio looks higher density, but the reality is all of these schematics are intended to show porous structures with similar density. We added a short explanation in the legend of Scheme 1 as “Note that all of the monolithic aerogels with these structures have the same bulk density.”

5. The high transparency is one key selling point: please also show photographs of aerogels on a black background and as a "window", i.e. not touching the background but in front of some sort of scenery or person. This gives a better impression of the real-world optical quality.

Reply: In light of the reviewer’s comment, we added a photo of one of the samples PMSQ-P94 in front of outdoor scenery to show the low-haze transparency, in Supplementary Fig. 14. Additional texts in the corresponding parts have also been included in page 18 in the revised manuscript as; “Because of the toughness, PMSQ-P94 was able to be prepared in a larger scale, and then the thermal conductivity was measured. The thermal conductivity of PMSQ-P94 is 14.5 mW m⁻¹ K⁻¹, which is comparable with conventional silica aerogels and low enough for highly insulating window applications. Therefore, as

demonstrated in Supplementary Fig. 14, the highly transparent and thermally insulating aerogel, PMSQ-P94, has good potential to be used as a window material because we can clearly see what is behind the aerogel.”

Supplementary Figure 14. A photograph of PMSQ-P94 taken outdoor to show the potential as a window insulating material. The scenery behind the aerogel can be seen clearly because of the low haze of the sample.

6. Data presentation is not always friendly to the color-blind.

Reply: We rechecked all the data and revised the graph presentations using solid/broken lines as well as colors, for better recognition by the color-blind, or those who read the hardcopies printed in black and white. We updated Figs. 2a, 3b, 4b,c, 5a. In addition, we revised the tabulated data in Figs. 3d, 5d and 6b to show the sample names instead of the data lines to improve the legibility.

7. Much is discussed on skeletal ratio versus bendability etc., but no correlations are shown and data are not tabulated close together. Please add e.g. skeletal ratios to Figure 4d (and also add sample names there!)

Reply: We updated Fig. 3 (old Fig. 4, shown above) by adding the definition of skeletal ratio in part c and tabulated data of the selected samples in part d. All the data are summarized in Supplementary Table 3.

Please note that we changed the discussions related to the structural parameters, because it is difficult to unambiguously define the porous structure from the structural data. In particular in the original manuscript, the two contrastive terms “fibrous skeletons” and “particle aggregated structure” were used, but we believe that the structures cannot be defined as one by one. We therefore revised such expressions to high (low) skeletal ratio structure, with supportive uses of the terms “fiber-like” (“particle aggregation-like”). In addition, we redefined the skeletal ratio parameter by (length of the skeleton)/(thickness of the skeleton). We believe the revised definition, in aspect ratio of the fiber-like parts, is more reasonable to evaluate the porous structure and make correlations with mechanical properties. Detailed definition is presented in Fig. 3c and the results are listed in Fig. 3d, 5d, and Supplementary Table 3. Additional texts in the corresponding parts have also been included in the revised manuscript.

8. I understand that 3 point bending is the main improvement here, but please also show compression data

(including compression, decompression cycles), as such data is more common and provides an easy comparison of performance to other studies.

Reply: We are sorry we did not show the results of compression tests. We added the compression-decompression cycle data in Supplementary Fig. 6 and 7, in which the prepared samples (data are shown for PMSQ-F127 and PMSQ-P105) are compressible to 50% without break and show spring-back to 80%. After relaxation at 110 °C for two hours, the PMSQ-F127 showed the nearly perfect spring-back. Additional texts in the corresponding parts have also been included in page 11 in the revised manuscript as; “Similarly, residual strains were observed on specimens of PMSQ-F127 and -P105 after 50% uniaxial compression test (Supplementary Fig. 6). In this case, however, the specimens recover nearly to their original state by heating at 110 °C for a few hours (Supplementary Fig. 7). These results indicate that both PMSQ-F127 and -P105 are viscoelastic.”

Supplementary Figure 6. Stress–strain curves obtained by a uniaxial compression–decompression test with 50% strain on PMSQ-F127 and -P105. There are some residual strains on both PMSQ-F127 and -P105 after the test.

Supplementary Figure 7. Photographs of viscoelastic behavior of PMSQ-F127 during a uniaxial compression–decompression test with 50% strain. The specimen is cylindrical, ca. 10 mm diameter and height. The compressed specimen is approximately 80% of the pre-test height (from 10.3 to 8.60 mm). However, the compressed specimen recovers its original shape (10.0 mm height) by heating at 110 °C for 2 hours.

9. Also for 3 point bending, it would be good to have some strain recovery data.

Reply: We are sorry we did not show the strain recovery data of three-point bending tests. We performed a cyclic bending test using a PMSQ-F127 specimen and the result is shown in Supplementary Fig. 5. The

specimen was bent just before breaking (Supplementary Fig. 5b), and when the load was removed, it returned to a slightly deformed state (Supplementary Fig. 5c). After this test, the specimen was heated at 120 °C overnight, but did not show a complete recovery, which is different from the result of compression test showing nearly complete recovery after annealing at 110 °C for two hours as described above. This explanation is added in the revised manuscript in pages 10–11.

Supplementary Figure 5. Deformation and recovery of PMSQ-F127 in a single three-point bending cycle test with 60 mm span. (a) A stress–strain curve obtained by bending to 15% strain. (b, c) Appearance of PMSQ-F127 at (b) 15% bending strain and (c) the end of the cycle test.

Added in the text as; “The stress–strain recovery behavior of a cyclic three-point bending test on PMSQ-F127 is shown in Supplementary Fig. 5. The specimen was bent just before breaking (Supplementary Fig. 5b), and when the load was removed, it returned to a slightly deformed state (Supplementary Fig. 5c). After this test, the specimen was heated at 120 °C overnight, but did not show a complete recovery.”

REVIEWERS' COMMENTS

Reviewer #2 (Remarks to the Author):

The revision has addressed my previous comments. I can recommend the publication of this manuscript in the current form.

Reviewer #3 (Remarks to the Author):

The manuscript has been revised thoughtfully and, in my opinion, is now suitable for publication. It is an exciting and excellent piece of aerogel research, with impressive properties.